# Ubiquitylome study identifies increased histone 2A ubiquitylation as an evolutionarily conserved aging biomarker

Lu Yang[1,2,7], Zaijun Ma[1,2,7], Han Wang[1,2], Kongyan Niu[1], Ye Cao[1,2], Le Sun[1,2], Yang Geng[1], Bo Yang[1,3], Feng Gao[4], Zuolong Chen [4], Zhen Wu[5], Qingqing Li[5], Yong Shen[4], Xumin Zhang [5], Hong Jiang[1], Yelin Chen[1], Rui Liu[6], Nan Liu[1] & Yaoyang Zhang [1]

The long-lived proteome constitutes a pool of exceptionally stable proteins with limited turnover. Previous studies on ubiquitin-mediated protein degradation primarily focused on relatively short-lived proteins; how ubiquitylation modifies the long-lived proteome and its regulatory effect on adult lifespan is unclear. Here we profile the age-dependent dynamics of long-lived proteomes in *Drosophila* by mass spectrometry using stable isotope switching coupled with antibody-enriched ubiquitylome analysis. Our data describe landscapes of long-lived proteins in somatic and reproductive tissues of *Drosophila* during adult lifespan, and reveal a preferential ubiquitylation of older long-lived proteins. We identify an age-modulated increase of ubiquitylation on long-lived histone 2A protein in *Drosophila*, which is evolutionarily conserved in mouse, monkey, and human. A reduction of ubiquitylated histone 2A in mutant flies is associated with longevity and healthy lifespan. Together, our data reveal an evolutionarily conserved biomarker of aging that links epigenetic modulation of the long-lived histone protein to lifespan.

[1] Interdisciplinary Research Center on Biology and Chemistry, Shanghai Institute of Organic Chemistry, Chinese Academy of Sciences, 26 Qiuyue Rd., Pudong, Shanghai 201210, China. [2] University of Chinese Academy of Sciences, Beijing 100049, China. [3] Shanghai Key Laboratory of Regulatory Biology, Institute of Biomedical Sciences, School of Life Sciences, East China Normal University, Shanghai 200241, China. [4] Neurodegenerative Disorder Research Center, University of Science and Technology of China, No.96, JinZhai Road Baohe District, Hefei, Anhui 230026, China. [5] State Key Laboratory of Genetic Engineering, Department of Biochemistry, School of Life Sciences, Fudan University, Shanghai 200438, China. [6] Singlera Genomics, 781 Cailun Road, Rm 1208, Pudong, Shanghai 201203, China. [7]These authors contributed equally: Lu Yang, Zaijun Ma Correspondence and requests for materials should be addressed to N.L. (email: liunan@sioc.ac.cn) or to Y.Z. (email: zyy@sioc.ac.cn)

Aging and many age-associated diseases are linked to functional decline and damages to the proteome[1,2]. Protein turnover can ameliorate such damages by degrading the impaired proteins, which can then be replenished by new protein synthesis[3]. Studies that measure protein turnover rates in different species have indicated a varied protein lifetime ranging from a few minutes to even years[4–7]. Intriguingly, recent studies have identified selected pools of long-lived proteins with limited turnover, which may persist for months or even years during adult life[8,9]. How the aging proteome, especially long-lived proteins, is modified and maintained during adult lifespan remain to be some of the most critical questions on aging that are not well-understood.

Ubiquitylation is a canonical post-translational modification (PTM) that involves the covalent attachment of ubiquitin, a polypeptide of 76 amino acids, to a substrate protein[10–14]. K48 ubiquitylation marks damaged or misfolded proteins for subsequent proteasomal degradation by the ubiquitin-proteasome system (UPS)[15–17]. In addition, ubiquitylation by other K residues, such as K63, mediates intracellular signaling events including mitochondrial protein turnover through autophagy[18–21], protein subcellular localization[22,23], and transcriptional regulation[24–27]. However, the manner by which ubiquitylation modifies the aging proteome and its regulatory effect on adult lifespan have not been investigated.

Growing evidence associates aging with epigenetic, transcriptional, and proteomic alterations. However, how these processes contribute to adult physiology and aging is only beginning to be revealed. For long-lived histone proteins, PTM through methylation and acetylation constitutes age-modulated epigenetic changes that profoundly impacts adult lifespan[28–30]. In particular, *Sce* and *Su(z)2* are components of the Polycomb Repressive Complex 1 (PRC1) that specifically modulates the ubiquitylation of histone proteins[31–38]. However, it is unclear how histone ubiquitylation is modulated with age and whether this modulation could impact adult lifespan.

Here we systematically analyze the dynamic changes in the proteomic turnover and ubiquitylation during aging with *Drosophila* as a model using quantitative mass spectrometry. Our analysis reveals an age-modulated ubiquitylation on long-lived histone 2A protein (ubH2A) in *Drosophila*, which is evolutionarily conserved in mouse, monkey, and human. Furthermore, we show that a reduction of ubH2A in mutant flies is associated with extended lifespan and enhanced ability to handle oxidative stress. Together, our data reveal an evolutionarily conserved biomarker of aging that links epigenetic modulation to adult lifespan.

## Results

**Isotope labeling strategy reveals long-lived proteomes**. To interrogate long-lived proteome, we adapted the pulse-$^{15}$N method, originally known as stable isotope labeling in mammals (SILAM), in *Drosophila* (Fig. 1a). Briefly, flies were fed with $^{15}$N-labeled diet started in utero. For young animals at 5d post-eclosion, fly diet was switched from $^{15}$N-labeled heavy form to natural $^{14}$N-labeled light form, such that newly synthesized proteins during adult lifespan could incorporate $^{14}$N, allowing the separation of the newer proteome from older proteome by mass spectrometry. Using this method, we characterized the *Drosophila* age-dependent changes in proteomes. Head and muscle of non-dividing somatic tissues, and testis of mitotically active germ-line were isolated from animals of 5d, 30d and 60d of age. Mass spectrometry analysis of 5d old animals fed with $^{15}$N-diet from in utero showed that $^{15}$N-labeled peptides accounted for more than 99.7% of the total peptides analyzed (Supplementary Fig. 1a), suggesting a near-complete labeling efficiency. A total number of

3074, 1903, 3034 proteins were quantitatively analyzed from all three aging time points in head, muscle and testis, respectively (Supplementary Fig. 1b and Supplementary Data 1–3). The turnover of proteomes during aging is demonstrated with the synthesis of new proteins and the decline in the older proteomes as shown by the reduction in the proportion of proteins with heavy $^{15}$N labels (Fig. 1b). The relative ratios of $^{14}$N and $^{15}$N of individual proteins between young and aged animals differed dramatically, thus demonstrating their differential turnover rates (Fig. 1b).

To map the proteome of long-lived proteins (LLP) in *Drosophila*, we first defined long-lived proteins as those with at least 10% of $^{15}$N-labeled forms remained at the age of day 60 compared to that of near 100% at 5d. Notably, this criterion was more stringent than previous studies in rat LLPs where 5% cutoff was used[9]. Our analysis showed that muscle proteome had the highest proportion of long-lived proteins, with 1502 out of 1903 (78.9%) proteins being LLPs; in comparison, 1715 out of 3074 (55.8%) proteins in the head proteome were LLPs (Supplementary Fig. 1b, c). In contrast, testis of *Drosophila* germline, where cells are mitotically active, had only 928 LLPs, accounting for 30.6% of identified testicular proteins. Gene Ontolgoy (GO) analysis found that LLPs shared by all three tissues were associated with energy generation (Supplementary Fig. 1d). While head-specific LLPs were involved in neurotransmission, LLPs in muscle and testis were preferentially enriched in cell cycle and metabolic processes (Supplementary Fig. 1e). Combined, our study of *Drosophila* aging proteomes by pulse-$^{15}$N reveals the roles of LLPs in regulating energy metabolism and tissue specific function such as neurotransmission in the heads.

We next clustered proteins according to the proportion of $^{15}$N using the Fuzzy C-means method. This analysis identified a cluster of proteins that retained more than 70% of $^{15}$N forms at 60d, thus representing the extremely long-lived proteins (ELLPs) in *Drosophila* (Fig. 1c). Altogether, our study cataloged 276, 503, and 306 ELLPs in head, muscle, and testis, respectively (Fig. 1c). Among 16 ELLPs shared by all three tissues, 3 were histone proteins, and 4 were laminin proteins (Supplementary Fig. 1f). Moreover, GO analysis of 149 ELLPs shared by both head and muscle indicated that nucleosome assembly was the most significantly enriched category ($p = 5.2$e-9, Fisher's exact test) (Fig. 1d). Strikingly, all previously reported long-lived rat histone proteins were found in this category, demonstrating the fact that histone proteins as ELLPs were also highly evolutionarily conserved[8,9](Fig. 1e). In addition, we identified an additional cohort of structure proteins as ELLPs, such as actin, tropomyosin, and paramyosin, in *Drosophila*.

**Ubiquitylation is over-represented in older proteins**. Given that ubiquitylation plays a critical role in regulating protein turnover, we interrogated how ubiquitylation might be involved in aging proteome. To address this question, we combined the pulse-$^{15}$N with antibody-enriched ubiquitylation analysis to allow quantitative measurements of ubiquitylome on both old and newly synthesized proteins. In fly heads from 60d old animals, 5317 unique ubiquitylation sites in 2068 proteins were identified, in which 2734 unique ubiquitylation sites in 988 proteins were quantified (Supplementary Data 4). While 451 quantified proteins contained only one ubiquitylation site, 537 quantified proteins were found to have more than one sites including 37 quantified proteins harboring even more than 10 modifications (Supplementary Fig. 2).

We next examined whether ubiquitylation might differentially modify old versus newly synthesized proteins. Analysis of peptide-spectrum matches (PSMs) from head tissues

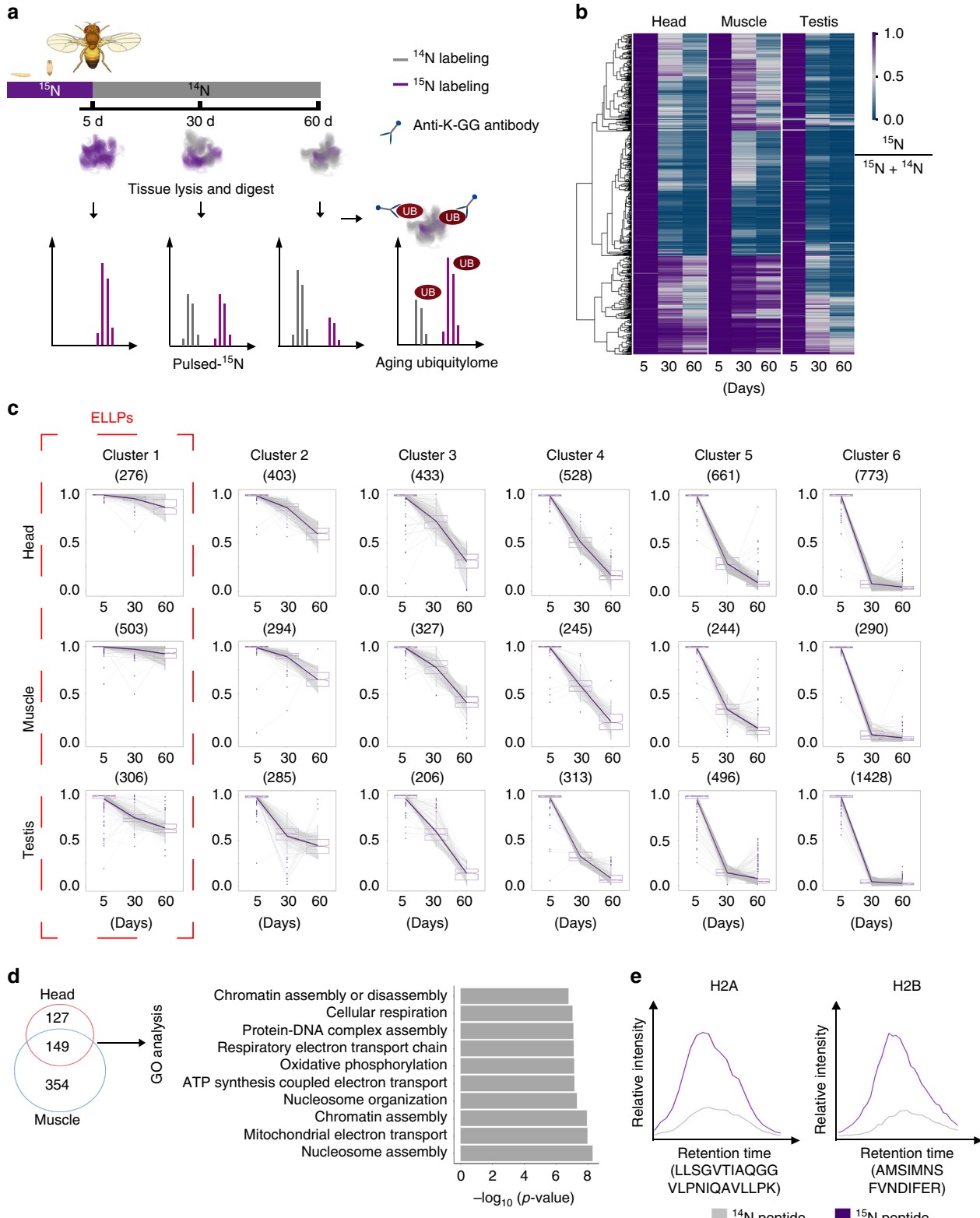

determined that while $^{15}$N-labeled old forms accounted for 28% of the entire proteome at 60d, they represented 59% of the ubiquitylated proteome (Fig. 2a). Thus, the older proteomes showed increased modification by ubiquitylation. Consistently, the majority of ubiquitylated proteins (~64%) displayed a higher percentage between $^{15}$N and $^{14}$N forms compared to their normal ratio in the aging head proteome as revealed by the above pulse-$^{15}$N experiments ($p = 2.2$e-16, Mann–Whitney-

Wilcoxon test) (Fig. 2b). Thus, ubiquitylation of individual proteins were also comparatively increased in the older proteins than that of their young counterparts. We further extended our analysis into muscle samples, and found a similar preference of ubiquitylation on old proteins. Our results showed that 68 and 84% of the total PSMs were identified as $^{15}$N-labeled in proteomic and ubiquitylomic datasets, respectively (Supplementary Fig. 3).

**Fig. 1** Long-lived proteome analysis during *Drosophila* aging. **a** Experimental scheme of metabolic labeling in *Drosophila*. Flies were first fully labeled with [15]N, and then switched to normal [14]N food at 5d. Samples were collected at 5d, 30d, and 60d of age. Proteomes of three tissues with indicated ages, and the ubiquitylome of 60d old head and muscle tissues were analyzed. The proportion of [15]N signal represents protein turnover rate. **b** [15]N proportions ([15]N %) for each protein in different tissues and different aging time point. Proteins quantified during aging were compared across tissue types. A high [15]N % value at aging time point represents a slow turnover rate. The protein identifiers were used for comparison among multiple datasets in this study. **c** Clustering of proteins according to their [15]N proportion changes during aging. Proteins in cluster 1 were characterized with the slowest turnover rates thus defined as extremely long-lived proteins (ELLPs). The median values are marked at the center. The boxes extend from the 25th to 75th percentiles. The upper or lower whiskers extend from the hinge to the largest or smallest value no further than 1.5 times the interquartile range (IQR).**d** Venn diagram and GO analysis of ELLPs in head and muscle. 149 ELLPs were found in both head and muscle, two post-mitotic tissues, and then subjected to functional analysis, with "nucleosome assembly" as the most significant hit ($p$ = 5.2e-9, Fisher's exact test). **e** H2A and H2B are ELLPs. H2A and H2B belonging to "nucleosome assembly" category were identified with a dramatic enrichment of [15]N signals at day 60. The extracted ion chromatographs (XICs) of the paired [15]N and [14]N peaks from two representative peptides were compared

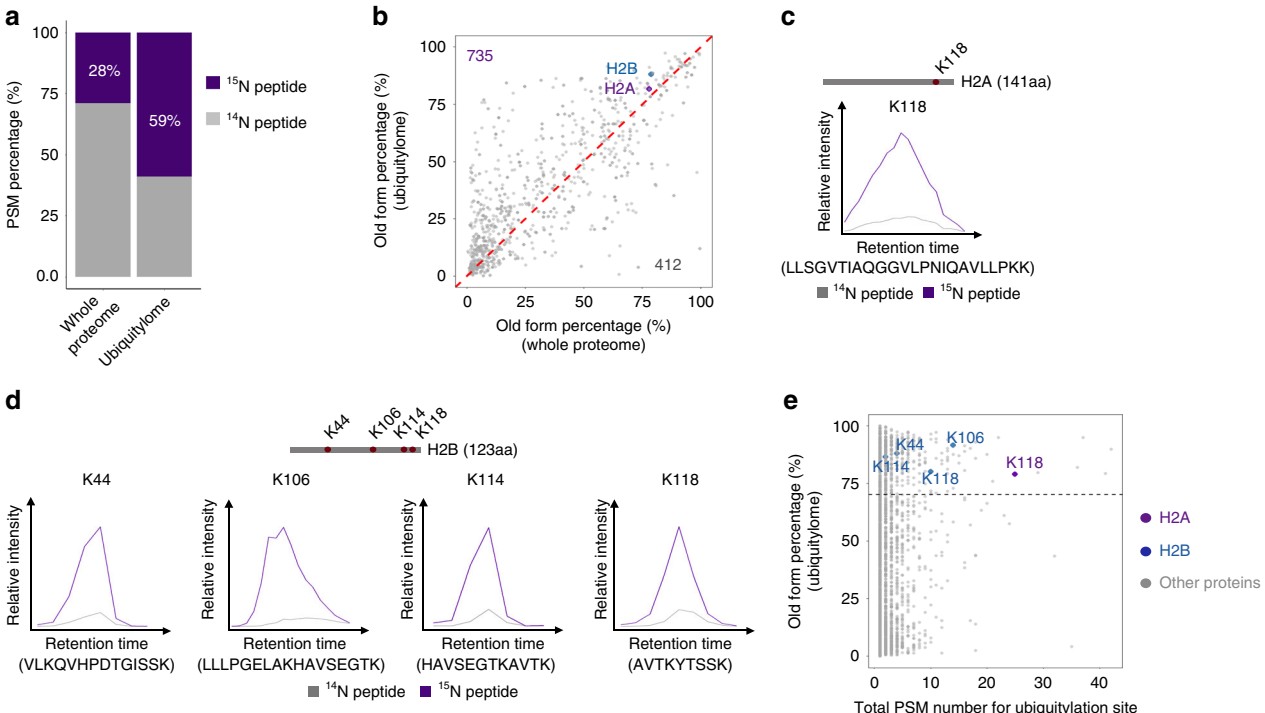

**Fig. 2** Ubiquitylome analysis of the aging proteome in fly heads. **a** Proportions of [15]N-PSMs in total PSMs. While [15]N-labeled old forms accounted for 28% of the entire proteome at 60d, they represented 59% of the ubiquitylated proteome, suggesting increased ubiquitylation in older proteome. **b** Quantifications of [15]N signals in total proteins and ubiquitylated proteins. A scatter plot of [15]N ratio confirms a majority of the proteins have higher [15]N proportion in ubiquitylome than that of the normal aging proteome at 60d of age ($p$ = 2.2e-16, Mann–Whitney-Wilcoxon test). **c, d** Ubiquitylated histone proteins are featured by higher [15]N ratios, including H2A (**c**) and H2B (**d**). The XICs of the paired [15]N and [14]N peaks were compared. **e** Quantification of [15]N proportions and abundance of ubiquitylated sites. H2A has high-abundance in the level of ubiquitylation, as assessed by the PSM counts

**ubH2A is a highly conserved aging biomarker**. We next characterized a group of individual long-lived proteins with significant ubiquitylation. This group included myosin heavy chain, voltage-dependent anion-selective channel, sodium/potassium transporting ATPase, and notably histone proteins (Supplementary Table 1). Interestingly, we noted a significant enrichment of ubquitylated histone 2A at lysine 118 (ubH2A) and multi-ubiquitylated histone 2B (Fig. 2c–e).

Since 119 K of H2A in mammals (K118 in fly H2A) is a conserved site for histone mono-ubiquitylation[36,39], we asked whether ubiquitylation of H2A might be modulated with age in different species (Fig. 3a). We first examined the levels of ubH2A by western blotting and revealed an increase of ubH2A in aged *Drosophila* (Fig. 3b), which was independently confirmed by MS analysis (Fig. 3c). By using western blotting and MS analysis, we determined increased accumulations of ubH2A in the brain and heart of aged mice compared to that of young animals (Fig. 3d–g). Moreover, the abundance of ubH2A was substantially increased

with age in the parietal lobes of non-human primate rhesus macaque (Fig. 3h–i) and prefrontal cortex of human (Fig. 3j). Combined, these data demonstrate that age-modulated increase of ubH2A is a highly conserved biomarker of aging.

**Reducing ubH2A couples with extended lifespan in *Drosophila***. To understand the biological significance of ubiquitin-modified long-lived proteins, we chose to explore how H2A ubiquitylation could impact adult lifespan in *Drosophila*. As noted, studies have implicated *Sce* and *Su(z)2* in modulation of H2A ubiquitylation[31–35]. We generated site-specific null mutants of *Sce*[c244] and *Su(z)2*[c433] using CRISPR/Cas9 mutagenesis[40] (Supplementary Fig. 4a, b). Given homozygous *Sce*[c244] or *Su(z)2*[c433] had a pre-adult lethality, we examined adult phenotypes by using the heterozygous mutants. qRT-PCR analysis confirmed that heterozygous mutants had reduced gene expression (Supplementary Fig. 4c). Fly feeding assay found no significant difference in food

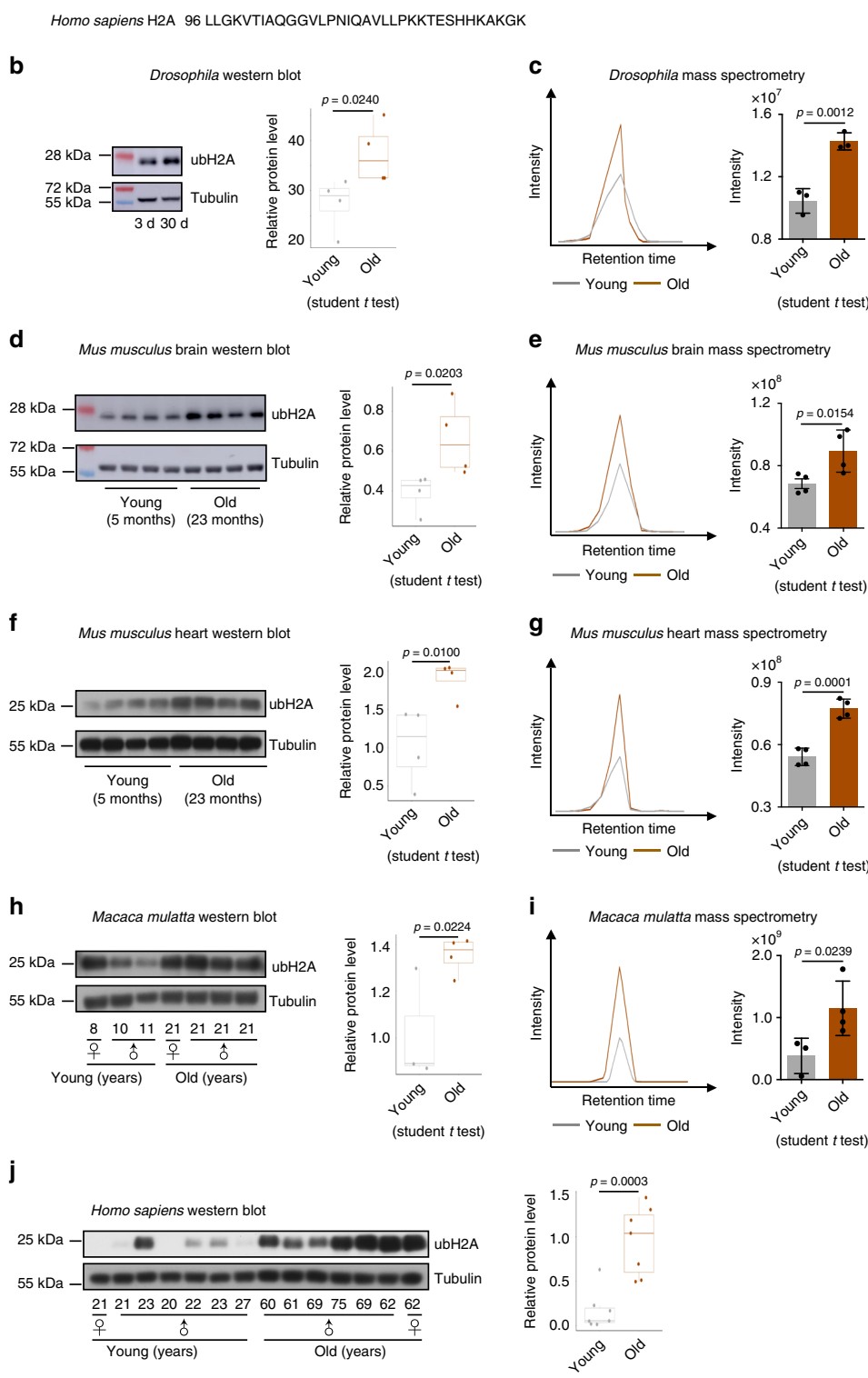

intake between WT and mutant flies (Supplementary Fig. 4d). Lifespan analysis demonstrated that animals deficient in $Sce^{c244}$ showed a reduction in the levels of ubH2A and lived longer than WT (Fig. 4a, b). Consistently, $Su(z)2^{c433}$ mutants also had extended lifespan and reduced ubH2A (Fig. 4a, b). Since the effects of epigenetic genes may have functional redundancy[41], we

further analyzed adult survival using $Sce^{c244}$, $Su(z)2^{c433}$ trans-heterozygous double mutants. Remarkably, double mutant male flies exhibited more striking effects on life-extension than single mutants (Fig. 4a). Consistently, female flies deficient in $Sce$ and $Su(z)2$ also manifested extended longevity and reduced levels of ubH2A (Supplementary Fig. 5). Independent lifespan

**Fig. 3** H2A mono-ubiquitylation increases with age in *Drosophila*, mouse, monkey, and human. **a** Alignment of amino acid sequence showing striking conservation of H2A in *Drosophila*, mouse, monkey, and human. **b, d, f, h, j** Western blotting (left) and quantification (right) show an evolutionarily conserved increase of ubH2A with age. Western blotting was from head tissues of 3d and 30d old male flies ($n = 4$ independent biological repeats) (**b**), brain and heart tissues of 5 months ($n = 4$) and 23 months ($n = 4$) old male mice (**d, f**), parietal lobes of young ($n = 3$) and old ($n = 4$) rhesus macaque (**h**), and prefrontal cortex of brain tissues of young ($n = 7$) and old ($n = 7$) male or female humans (**j**). The median values are marked at the center. The boxes extend from the 25th to 75th percentiles. The *p*-values are calculated by using the student *t* test. **c, e, g, i** MS quantifications confirm the increase of ubH2A with age in *Drosophila* ($n = 3$ independent biological repeats)(**c**), mouse brain ($n = 4$ for both young and old groups) (**e**), mouse heart ($n = 4$ for both young and old groups) (**g**), and monkey ($n = 3$ for young, $n = 4$ for old) (**i**). The ubH2A was measured by PRM-based targeted mass spectrometry method. Representative XICs of the y3 ion were shown. The quantification was done by using the summed intensities of y3, y4, y5, y6, and y7 ions. Data are shown as mean ± s.d. The *p*-values are calculated by using the student *t* test

experiments were performed with similar phenotypes (Supplementary Fig. 5b and 6). Moreover, detailed examination using western blotting confirmed that only ubH2A was selectively reduced in the double mutants, while other histone markings remained unchanged (Supplementary Fig. 7). Combined, these data indicate that a reduction of ubH2A levels is coupled with adult longevity.

Histone modification modulates the chromatin accessibility to control gene expression[42,43], and specifically, upregulation in the level of ubH2A has been implicated in regulating gene repression[39]. We then asked whether transcriptional change of particular genes might account for ubH2A-dependent regulation on aging. For this and subsequent experiments, we used male flies of *Sce*[c244], *Su(z)2*[c433] trans-heterozygous double mutants that gave rise to the stronger life-extension phenotype than the single mutants. Using dissected head tissues, we generated RNA-seq datasets for polyA-selected mRNAs (Fig. 4c). Pathway analysis revealed that glutathione-related pathway and unfolded protein response among others were significantly upregulated in double mutants as compared to age-matched WT animals (Fig. 4c). Specifically, the increase of *GSTO1* has been previously implicated in oxidative stress response relevant to lifespan modulation in *C. elegans*[44]. qRT-PCR analysis confirmed that *GSTO1* transcription had two-fold increase in double mutants relative to WT (Fig. 4d). We thus examined glutathione, an important indicator of the cellular redox state. Our data showed that GSH/(GSH + GSSG) ratio in mutants was significantly higher than that in WT (Fig. 4e), suggesting improved cellular redox potential in *Sce*[c244], *Su(z)2*[c433] double mutants. Since some other genes also displayed altered expression between WT and age-matched mutants, we could not rule out the possibility that additional mechanisms might be involved.

We then asked how this ability of enhanced redox potential in *Sce*[c244] single mutants and *Sce*[c244], *Su(z)2*[c433] double mutants was translated into fitness and stress resistance in aging. To illuminate this, we characterized age-related phenotypes. Analysis of climbing ability demonstrated that mutants had better locomotion than that of age-matched WT (Fig. 5a). To examine the sensitivity to oxidative stress, we utilized hydrogen peroxide ($H_2O_2$), a source of reactive oxygen species (ROS). We observed substantially enhanced resistance to oxidation in *Sce*[c244] single mutants and *Sce*[c244], *Su(z)2*[c433] double mutants than WT (Fig. 5b). Independent oxidative stress test was performed with a similar effect (Supplementary Fig. 8). Taken together, these data implicate that a reduction of ubH2A by gene mutation couples longevity with enhanced ability to handle stress, a characteristic associated with improved adult fitness.

## Discussion

A prominent feature of aging is a gradual decline in protein homeostasis network, particularly those of long-lived proteins that cannot be efficiently removed by the protein quality control system and replenished by new protein synthesis. As a consequence, long-lived proteins may be subject to chemical modification and damage leading to altered or loss of function over adult lifespan, and their identification is thus important for our understanding of the aging process. Here, we provide a comprehensive analysis of long-lived proteins in *Drosophila* somatic and reproductive tissues. Moreover, we profile the ubiquitin-modified long-lived proteome and demonstrate an increase in the levels of ubH2A as biomarker of aging. While H2A is a long-lived protein, our data showed that in aging fly heads (60 days), the [15]N-labeled old forms of ubH2A were only minimally increased (82%) compared to 78% of total H2A protein that was composed of old forms. Thus, our data do not suggest that ubiquitylation of H2A contributes to its stability in aging context. Our study suggests that ubiquitylation on histone protein may be linked to epigenetic and chromatin modifications during aging which should be examined by further studies.

The recent reports of long-lived proteins in rat indicate that proteins with extraordinary lifespan are more prevalent than previously appreciated[9]. Our quantitative assessment shows that fly heads and muscles, which are primarily composed of post-mitotic cells in the adult stage, contain a significant amount of proteins with slow or limited turnover in adult life. In contrast, the extent of long-lived proteins becomes comparatively lower in the fly germline such as testis that is composed of mitotically active cells compared to that of somatic tissues, suggesting that cell division coupled with new protein synthesis might contribute to replenishing the proteome. Given the fact that fly has relatively shortened lifespan, it is possible that the effect is more dominant in the *Drosophila* model than other longer living animals. However, many *Drosophila* ELLPs are in good agreement with those found in the rat model, such as nuclear and structure proteins[9].

Although the long-lived proteome is now well-recognized in multiple model organisms, how these proteins are maintained and modified during aging remain poorly understood. For long-lived Nup93 and Nup153, it has been shown that carbonyl groups can be added to the proteins[45]. Moreover, PARylation is found to be markedly increased in aged mouse liver and muscle[46]. Our study further extends the scope by profiling the interface between ubiquitylation and aging proteome. Importantly, we provide evidence to show a significant preference for old proteins to be ubiquitylated compared to their newly synthesized counterparts. These data may implicate an age-dependent decline in UPS, which aligns with a previous report in flies[47], and protein turnover during aging as which would manifest a gradual accumulation of ubiquitin-modified proteins. Interestingly, whereas ubiquitin-modified protein turnover affects the stability of individual proteins, age-modulated ubiquitylation on histone may lead to global and site-specific chromatin remodeling, which in turn may trigger a much broader effect in gene expression linked to altered cellular fitness. It has been shown that, in flies and humans, ubH2A binds to the promoter regions of Polycomb-regulated genes, including the Hox genes[32,48]. Thus, it is possible

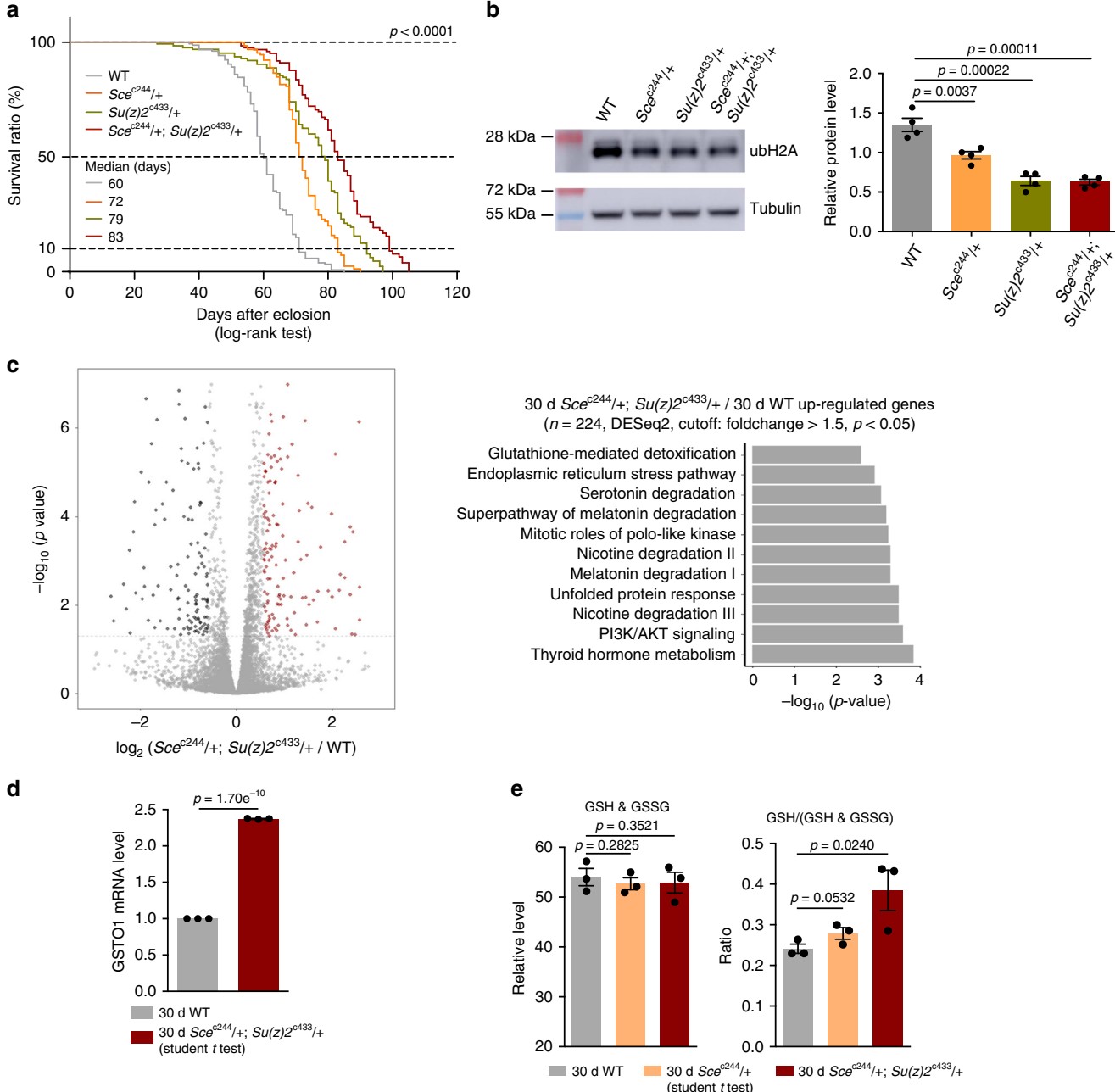

**Fig. 4** Reducing ubH2A couples with extended adult lifespan in *Drosophila*. **a** $Sce^{c244}/+$ or $Su(z)2^{c433}/+$ single mutants and $Sce^{c244}/+$ ; $Su(z)2^{c433}/+$ double mutants have extended lifespan. All lifespan data of this study were acquired in at least two replicates. (For lifespan: 25 °C; $n > 100$ male flies per genotype for curve; log-rank test). Genotypes (fly total number (death / censored)): WT: 5905 (156 (154 / 2)). $Sce^{c244}/+$ (128 (113 / 15)). $Su(z)2^{c433}/+$ (126 (111 / 15)). $Sce^{c244}/+$ ; $Su(z)2^{c433}/+$ (144 (103 / 41)). **b** ubH2A is significantly decreased in $Sce^{c244}/+$, $Su(z)2^{c433}/+$, and $Sce^{c244}/+$ ; $Su(z)2^{c433}/+$ flies ($n = 4$ independent biological repeats for ubH2A quantification) (mean ± s.e.m.; student *t* test). Protein was from head tissues of 30d old male heads. **c** Volcano plot shows that genes are differentially expressed in $Sce^{c244}/+$ ; $Su(z)2^{c433}/+$ double mutants compared to WT (left panel). Differential expression analysis was performed by DESeq2 from 3 biological replicates. 224 genes were found to be upregulated (Fold change >1.5, $p <$ 0.05). Ingenuity pathway analysis (right panel) shows that "glutathione-mediated detoxification" was significantly enriched for genes upregulated in the $Sce^{c244}/+$ ; $Su(z)2^{c433}/+$ double mutant ($p = 0.0026$, Fisher's exact test). RNA-seq was from heads of 30d old male flies. Genotypes: WT: 5905. $Sce^{c244}/+$ ; $Su(z)2^{c433}/+$. **d** qRT-PCR analysis shows that *GSTO1* of the glutathione-mediated detoxification pathway becomes significantly increased in the $Sce^{c244}/+$ ; $Su(z)2^{c433}/+$ double mutants ($n = 3$ independent biological repeats for qRT-PCR) (mean ± s.e.m.; student *t* test). qRT-RNA was from heads of 30d old male flies. Genotypes as in **c**. **e** While GSH and GSSG levels remain unchanged (left panel), GSH/(GSH + GSSG) ratio (right panel), indicator of the cellular redox level, is higher in $Sce^{c244}/+$ single mutants and $Sce^{c244}/+$ ; $Su(z)2^{c433}/+$ double mutants than age-matched WT. Metabolic analysis was from heads of 30d old male flies. Genotypes: WT: 5905. $Sce^{c244}/+$. $Sce^{c244}/+$ ; $Su(z)2^{c433}/+$ ($n = 3$ independent biological repeats; student *t* test)

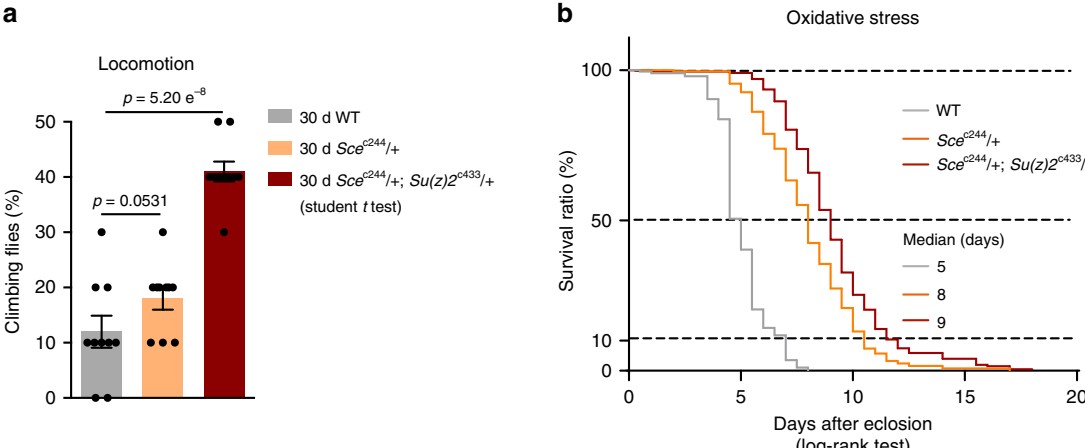

**Fig. 5** Reducing ubH2A couples with an effect to promote adult fitness. Analysis of adult phenotypes reveals that $Sce^{c244}/+$ single mutants and $Sce^{c244}/+$; $Su(z)2^{c433}/+$ double mutants have better climbing (**a**) and enhanced resistance to oxidation (**b**) than age-matched WT. (for climbing assay: mean ± s.e.m. of 10 biological repeats with 10 male flies for each repeats; student $t$ test; for oxidation tests: 25 °C, $n > 100$ male flies per genotype for curve, log-rank test). Genotypes (fly total number (death / censored)): WT: 5905 (196 (196 / 0)). $Sce^{c244}/+$ (245 (245 / 0)). $Sce^{c244}/+$; $Su(z)2^{c433}/+$ (202 (201 / 1))

that ubH2A is involved in transcriptional repression by introducing additional histone marking. Moreover, an emerging crosstalk between ubH2A and methylation on histone H3 at lysine 4 is also reported[49]. Though increased ubiquitylated H2B has been observed during yeast replicative aging[50], the biological significance of histone ubiquitylation on adult lifespan has not been interrogated. Our study indicates that a reduction of ubH2A by gene mutation promotes adult healthy lifespan in *Drosophila* by increasing the expression of genes in anti-oxidative pathway. However, additional work will be required to determine the epigenomic feature of ubH2A that occurs during aging. As noted, the levels of select histone markings, e.g., tri-methylated histone H3 at lysine 27 (H3K27me3), may exhibit increased or decreased change with age in different model organisms or tissue types[51,52]. Thus, identification of conserved aging biomarker in histone modification has potential implications as a common epigenetic mechanism operative in the aging process. Since the change of ubH2A defines an evolutionarily conserved aging biomarker in fly, mouse, monkey, and human, future investigation, including comparative analysis of ubH2A epigenome in the aging process may unveil additional common mechanisms that modulate age-associated physiological decline and disease.

## Methods

**Fly culture**. Flies were cultured in standard media at 25 °C with 60% humidity in a 12 h light and 12 h dark cycle. The standard *Drosophila* food was prepared using the recipe consisting of 36 g L$^{-1}$ sucrose, 38 g L$^{-1}$ maltose, 22.5 g L$^{-1}$ yeast, 5.4 g L$^{-1}$ agar, 60 g L$^{-1}$ maizena, 8.25 g L$^{-1}$ soybean flour, 0.9 g L$^{-1}$ sodium benzoate, 0.225 g L$^{-1}$ methyl-p-hydroxybenzoate, and 6.18 ml L$^{-1}$ propionic acid. The WT (control) line used was 5905 (FlyBase ID FBst0005905, w1118). All fly lines used in this study have been backcrossed with 5905 for five consecutive generations for a uniform genetic background, to assure that phenotypes were not associated with any variation in background.

**15N labeling of flies for long-lived proteins**. Three-day-old male and female flies were crossed on media consisting of 10% (w/v) $^{15}$N-labeled *Saccharomyces cerevisiae* (Silantes GmbH, Germany), 150 mM sucrose, 6 mM methylparaben, and 0.5‰ propionic acid. Flies were maintained in a 12 h light/12 h dark cycle at 25 °C and 60% relative humidity. The parent flies were removed after 5 days. After 11 days, the newborn flies were collected and transferred to a $^{15}$N diet for another 5-day culture. Then the flies were switched to a normal $^{14}$N food until 30d and 60d old of age. The long-lived proteomic profiling was conducted in three biological replicates, and the ubiquitylomic analysis was performed in duplicates.

**Sample preparation for LC-MS analysis**. Head, muscle and testis tissues were homogenized in ice-cold urea lysis buffer containing 8 M urea, 100 mM Tris-HCl, pH 8.5, 150 mm NaCl, and protease inhibitor (Pierce, USA). The homogenate was centrifuged using 14,000 × g for 10 min, and the protein concentration of the supernatant was determined using the BCA protein assay (Pierce, USA). Proteins were digested with trypsin by a standard protocol of filter-aided sample preparation[53].

**Basic reversed phase (RP) chromatography**. Off-line basic RP HPLC fractionation was performed on a XBridge BEH C$_{18}$ column using an Agilent 1260 HPLC system. 500 μg or 8 mg of peptide was fractionated by using a 72 min basic RP HPLC gradient for aging proteome or ubiquitylome analysis respectively. Upon sample injection, 72 fractions were collected into 1.5 ml Eppendorf tubes (Eppendorf, UK). The resulting small fractions were then pooled in a non-contiguous manner into 7–8 large fractions. The original fractions 1, 9, 17, 25, 33, 41, 49, 57 and 65 were combined to generate the first pooled fraction. The remaining seven fractions were created followed by a similar pooling strategy. Fractions were dried to completeness in a vacuum concentrator. Peptides from each fraction were subjected to LC-MS analysis or ubiquitylation enrichment using antibody recognizing K-GG remnant.

**Ubiquitylated-peptide enrichment**. The peptide mixture about 1 mg each fraction were dissolved in 200 μl IP buffer (50 mM MOPS pH 7.2, 10 mM sodium phosphate, 50 mM NaCl, and 0.3% NP40). Total 80 μl of K-GG ubiquitin remnant motif antibody bead conjugate (PTM-1104, PTM Biolabs, China) were washed three times with cold PBS, then were loaded into 8 homemade C$_8$ StageTips equally. Each fraction of re-suspended peptides, were add to C$_8$ StageTips and flow past beads by centrifuge force of 200 × g. IP buffer and IP buffer without NP40 were used to wash beads for 2 times separately to remove unspecific-binding peptides and following three times wash with H$_2$O by centrifuge force of 200 × g. To elute peptides, 40 μl of 0.2% trifluoroacetic acid (TFA) were loaded to the tips and centrifuge at 100 × g to collect peptides. The resulting peptides were dried to completeness.

**LC-MS/MS**. On-line LC-MS/MS analysis was performed on a tribrid Orbitrap fusion mass spectrometer coupled to a NanoLC-1000 HPLC system (Thermo Fisher Scientific, USA). The peptide mixture was separated by an in-house manufactured 15-cm fritless column packed with C$_{18}$ resin (1.9-μm, Dr. Maisch GmbH, Germany) at a flow rate of 300 nl/min. Mass spectra were acquired in a data-dependent mode with one full scan in the Orbitrap (m/z: 350–1800; resolution: 60000; AGC target value: $5 \times 10^5$), followed by MS$_2$ scan in the linear trap (32% normalized collision energy; AGC target value: 10000, maximal injection time: 50 ms or 200 ms for tissue lysate or ubiquitylation analysis, respectively). The dynamic exclusion for precursor selection was set as 60 s. For parallel reaction monitoring (PRM) measurement, precursor ions with m/z of 848.5143, 725.1071, and 838.5108 corresponding to triply charged ubiquitylated peptides of LLSGVTIAQGGVLP-NIQAVLLPKK (fly), VTIAQGGVLPNIQAVLLPKK (mouse), and LLGGVTIAQGGVLPNIQAVLLPKK (monkey), were selected for MS$_2$ analysis. The precursor peptides were chosen based on their high detection frequency in both our and other's shotgun data[54,55].

**Protein identification and quantification**. 60d aged flies' head, muscle and testis were used to identify long-lived proteins. Raw data of different fractions belong to one sample were searched together against Uniprot *Drosophila* database (download date: Feb 2016) using Integrated Proteomics Pipeline - IP2 (Integrated Proteomics Applications, http://www.integratedproteomics.com/)[56], with 20 ppm and 400 ppm were set for precursor and fragment mass tolerance respectively. Trypsin was set as the enzyme, and maximum allowed cleavage was 1 and 2 for tissue lysate or ubiquitylation-enriched sample. Static modification was carbamidomethyl (C) (+57.02 Da), while dynamic modification was oxidation of methionine (+15.99 Da) for all samples, and diGly of lysine (+114.04 Da) only for ubiquitylation-enriched samples. The protein false discovery rate (FDR) was controlled to be less than 1% for all datasets using a decoy protein database. The peptide and spectrum FDRs were calculated as ~ 0.3% for ubiquitylated peptide identification. $^{15}$N analysis was performed in parallel using the same parameters. $^{14}$N and $^{15}$N ratio of each peptide was calculated using Census[57,58]. Two peptides were required for protein quantification, while only one ubiquitylated peptide was required to report a ubiquitylated protein. For PRM analysis, the peptide samples derived from young and old animals with equal amount of starting total protein content were analyzed. The acquired MS data were analyzed by Skyline software[59]. The summed MS intensities of y3 to y7 ions, representing the five most intense peaks in MS$_2$, were used for quantification without further normalization.

**Western blotting assays for histone modification**. All animal procedures were reviewed and approved by the Institutional Animal Care and Use Committee at Chinese Academy of Sciences and are in accordance with the Guide for the Care and Use of Laboratory Animals of Chinese Academy of Sciences. The level of ubiquitylated histone H2A was analyzed by western blotting using dissected heads from 3d and 30d old flies. Male C57 mice were used at the ages of 5 months and 23 months, with 4 mice per group. Rhesus macaque parietal lobes were obtained from the brain bank of University of Science and Technology of China. Human prefrontal cortex from 2 females and 12 males were obtained from Chinese Brain Bank Center (CBBC), Hubei, China. Equal amount of tissues were lysed in RIPA buffer containing 50 mM Tris (pH 7.4), 150 mM NaCl, 1% NP-40, 0.25% sodium deoxycholate and protease inhibitor (Pierce, USA) with motor pestle. Protein mixture was centrifuged at 16000 g, for 10 min at 4 °C to remove undissolved ceratine, and then separated on a NuPAGE 12% Bis-Tris gel (GeneScript, China). The separated proteins were then transferred to a polyvinylidene fluoride membrane (Millipore, USA). The primary antibody was diluted 1:2000 (anti-ubH2A, #8240, rabbit host, CST, USA) for *Drosophila* head, mouse brain and heart, monkey brain, and human brain samples, with overnight incubation at 4 °C. The HRP-conjugated secondary anti-rabbit antibody (31460, Pierce, USA) was diluted 1:10000 and incubated for 1 h at room temperature. The following antibodies were used in western blots for H3 and H4 epigenetic markers: rabbit anti-H3K27me3 (1:1000, 07–449, millipore), rabbit anti-H3K27me2 (1:1000, ab24684, abcam), rabbit anti-H3K27ac (1:1000, ab4729, abcam), rabbit anti-H3K4me3 (1:1000, 07–473, millipore), rabbit anti-H3K4me2 (1:1000, 07–030, millipore), rabbit anti-H3K9me3 (1:1000, ab8898, abcam), rabbit anti-H3K9ac (1:1000, 06–942, millipore), rabbit anti-H3K36me3 (1:1000, ab9050, abcam), rabbit anti-H3K36ac (1:1000, 07–530, millipore), rabbit anti-H3K14ac (1:1000, 07–353, millipore), rabbit anti-H3K18ac (1:1000, ab1191, abcam), rabbit anti-H4K20me3 (1:1000, ab9053, abcam), rabbit anti-H4K20me (1:1000, ab9051, abcam), rabbit anti-H4K12ac (1:1000, 07–595, millipore), goat anti-H3 (1:1000, ab12079, abcam), and rabbit anti-H4 (1:1000, ab10158, abcam). The signal intensity was quantified by ImageQuant TL (GE Healthcare, UK). The relative abundance of histone modification was normalized to loading control, and statistical analysis was using student's *t* test (Source Data).

**CRISPR-Cas9**. CRISPR/Cas9 mutagenesis was performed as previously described[40]. Two sgRNA plasmids for target gene were injected into fly embryo. Single-fly-PCR assays were used to screen for mutants. To do this, single fly was homogenized in 50 μl squashing buffer (10 mM Tris buffer (pH 8.5), 25 mM NaCl, 1 mM EDTA, 200 μg/ml Proteinase K), then incubated at 37 °C for 30 min, followed by 95 °C, 10 min for inactivation. Screen primers for *Sce*$^{c244}$ (F:5'-GTCGCTGTACGAGCTGCAGCGCAAG-3'; R:5'-CGATGTGCCCGAATGACCCGGATCC-3') and *Su(z)2*$^{c433}$ (F:5'-CGATCCAACCACTGTGGATTACTG-3'; R: 5'-GTAGGCAGTACCTCATCCTTGTAG-3') deletion respectively. The virgin females carrying the deletion were backcrossed into WT (5905) male flies for five consecutive generations for a uniform genetic background, to mitigate background effects.

**Assay for fly feeding rate**. Fly feeding rate was examined as previously described[60]. For each genotype, 25 male files at 3d of age were transferred into an empty vial for 15 min. Then flies were equally divided into 5 vials with standard *Drosophila* food media containing 1% (w/v) FD&C Blue 1 (Aladdin, China) for 15 min. Flies were then homogenized in 1xPBS solution containing 1% Triton X-100, and the resulting homogenates were centrifuged at 10,000 × g for 5 min. The absorbance of the supernatant was recordead at 630 nm on a SYNERGY 2 microplate reader (BioTek).

**Lifespan**. Adult male flies were collected at the day of eclosion and maintained at 20 flies per vial at 25 °C with 60% humidity and a 12 h light/12 h dark cycle. Flies were transferred to new vials every other day and scored for survival (Source Data).

**GSH/GSSG measurement**. One hundred fly heads were harvested and immediately frozen in liquid nitrogen. Tissues were homogenized using the homogenizer (Precellys 24, Bertin Technologies, Germany) with 300 μl ddH$_2$O. Then frozen at −80 °C for 30 min and thawed at room temperature, followed by 5 cycles of 30 s on and 30 s off sonication. Lysates were centrifuged using 10,000 g at 4 °C for 5 min. Supernatant was transferred to 10 kDa molecular weight cutoff spin filter (Millipore, USA), followed by centrifugation using 10,000 g at 4 °C for 30 min to remove proteins. Assays were assembled according to the instruction of manufacturer (S0053, Beyotime Biotechnology, China). Absorbance at 412 nm were detected by the Multiskan GO (Thermo Fisher Scientific, USA).

**Climbing test**. Ten male flies were transferred to an empty vial for 30 min adaption in dark. Flies were tapped to the bottom of the vial; the percentage of flies abled to climb up to a mark at 2 cm from the bottom within 5 s was scored. Ten biological replicates were done for each genotype at given age.

**Oxidative stress**. Male flies for each genotype were used, with 20 flies per vial. Prior to the test, flies were pre-treated with 1% Agar media for 6 h before transferred to the media containing H$_2$O$_2$. Then, flies were transferred to vials containing a small piece of Kimwipe filter paper pre-soaked in 1.5 ml of 10% glucose and 2% H$_2$O$_2$ (Sinopharm Chemical Reagent, China). Dead flies were scored every 12 h, and the Prism software (Graphpad) was used to generate the survival curve and statistical analysis.

**RT-qPCR**. Fifty heads were dissected and homogenized in a 1.5 ml tube containing 1 ml of Trizol Reagent (Thermo Fisher Scientific, USA). RNA isolation was followed in accordance with manufacturer's instruction. RNA was re-suspended in DEPC-treated RNase free water (Thermo Fisher Scientific, USA). TURBO DNA free kit was used to remove residual DNA contamination according to manufacturer's instruction (Thermo Fisher Scientific, USA). 1 μg of total RNA was used for reverse transcription by random primers using SuperScript III First-strand synthesis system for RT-PCR (Thermo Fisher Scientific, USA). Analysis was performed using the QuantStudio 6 Flex real-time PCR system with SYBR selected master mix (Thermo Fisher Scientific, USA). The 2$^{-\Delta\Delta CT}$ method was used for quantification upon normalization to the RP49 gene as internal control. Primers for target gene *GSTO1*: F:5'-AGATCCCGTATCACAGCATCTAC-3'; R:5'-GCTCGTCCAGATATTCACAAATC-3'. Primers for *Sce*: F:5'-GATCCCAACTTCGACCTGCT-3'; R:5'-TGATGCCCTCGTTGATGGAG-3'. Primers for *Su(z)2*: F:5'-AGATTTCAAGGAGCAGCACGA-3'; R:5'-GCAGGTAAATAGTGGGAGGCA-3'. Primers for RP49: F: 5'-CCGCTTCAAGGGACAGTATCTG-3'; R:5'-ATCTCGCCGCAGTAAACGC-3'.

**PolyA-selected mRNA-seq**. Head tissues were homogenized in a 1.5 ml tube containing 1 ml of Trizol Reagent (Thermo Fisher Scientific, USA). RNA isolation was followed in accordance with manufacturer's instruction. RNA was re-suspended in DEPC-treated RNase free water (Invitrogen, USA). TURBO DNA free kit was used to remove residual DNA contamination according to manufacturer's instruction (Invitrogen, USA). A concentration of 1 μg of total RNA was used to generate sequencing library using VAHTS mRNA-seq v2 library Prep Kit for Illumina. The library quality was checked by Bioanalyzer 2100 (Agilent, USA). The quantification was performed by qRT-PCR with a reference to a standard library. The libraries were pooled together in equimolar amounts to a final 2 nM concentration. The normalized libraries were denatured with 0.1 M NaOH (Sigma, USA). Pooled denature libraries were sequenced on the illumina NextSeq 550 platforms with single end 100 bps. Sequencing reads were mapped to the reference genome dm6 with STAR2.3.0e[61] by default parameter. The read counts for each gene were calculated by HTSeq-0.5.4e[62] htseq-count with parameters "-m intersection-strict -s no". The count files were used as input to R package DESeq[63] for normalization and the differential expression genes were set at a *p*-value smaller than 0.05. (http://bioinfogp.cnb.csic.es/tools/venny/index.html).

**GO and pathway analysis**. Biological pathway analysis for *Drosophila* LLPs and ELLPs were performed with David[64,65]. Differential expression of mRNA genes was analyzed by ingenuity pathway analysis-IPA (QIAGEN, USA).

**Reporting summary**. Further information on research design is available in the Nature Research Reporting Summary linked to this article.

## Data availability

The MS data have been deposited to the ProteomeXchange Consortium (http://proteomecentral.proteomexchange.org) via the iProX partner repository under the accession number PXD013101 and with the project ID IPX0001204000. The Illumina RNA-seq data have been deposited in the Sequence Read Archive (SRA), using the NCBI

portal, under the BioProject accession number PRJNA453196 and SRA accession number SRP142472]. The original data for Fig. 3b-j, Fig. 4a, b, d, e, Fig. 5, Supplementary Fig. 4c, d, and Supplementary Fig. 5–8 are provided in the Source Data file. A Reporting Summary for this Article is available as a Supplementary Information file. All other data supporting the findings of this study are available from the corresponding authors upon reasonable request.

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

## Acknowledgements

We thank Prof. Junying Yuan for considerable support, advice on the experiments, and critical suggestions on the manuscript. Funding support was provided by the National Natural Science Foundation of China to Y.Z. (31671428, 31500665, 31530041), N.L. (91849109), Y.C. (31671044), and Y.S. (XDPB10), the 100 Talents Program of the Chinese Academy of Sciences to Y.Z., Shanghai Science and Technology Committee to Y. C. (17DZ1205402), and the National Program on Key Basic Research Project of China to Y.Z., Y.C., and N.L. (2016YFA0501900 and 2016YFA0501904).

## Author contributions

Y.Z. and N.L. conceived and supervised the study. L.Y., Z.M., H.W., K.N., Y.Ca. and L.S. conducted the experiments. L.Y., Z.M., N.L., and Y.Z. analyzed the data. N.L., Y.Z. and L.Y. wrote the manuscript. R.L. performed the mRNA-seq analysis. Y.G., B.Y., F.G., Z.C., Z.W., Q.L., Y.S., X.Z., H.J. and Y.Ch. provided experimental materials.

## Additional information

**Competing interests:** The authors declare no competing interests.

