## [Peer Review File · Nature Communications]

Reviewers' comments:

Reviewer #1 (Remarks to the Author):

The authors of this study used state-of-the-art proteomics technology to characterize long-lived proteins in *Drosophila*. Head, muscle, and testis samples were analyzed on adult days 5, 30, and 60. The flies were shifted from ¹⁵N-labeled food to unlabeled food on adult day 5. The authors analyzed the whole proteome and the ubiquitylated proteome in parallel and quantified for each peptide the percentage of remaining ¹⁵N labels. Proteins with a longer half-life will retain a high level of ¹⁵N labels. A total of 149 extremely long-lived proteins (ELLPs) were found by taking the overlap between the head and the muscle data. There is a good agreement between the fly ELLPs from this study and the mouse ELLPs from previous studies. Fly specific ELLPs were also identified. One very interesting discovery is that ubiquitylated peptides tend to retain a higher level of ¹⁵N labels than peptides not ubiquitylated, suggesting that ubiquitylated proteins tend to have a slower turnover rate or a longer half-life. This is unexpected but very intriguing, and it provides insight to the metabolism of the aging proteome. I think this is an important discovery and the data sets generated in this study will be an asset to the aging research community.

However, this manuscript is not without flaws. Two major ones:

First, that ubH2A is a highly conserved hallmark of aging is a premature conclusion. Yes, it is exciting to observe an increase of ubH2A in aged *Drosophila*, mice, and human, but to call it a hallmark of aging, one needs to examine it under different genetic or environmental conditions that alter lifespan. For example, between wild-type and long-lived mutant flies of the same age, is the ubH2A level higher in the former? Additionally, I like to point out that an increase of ubH2A with age is an incidental discovery. Although the WB experiments were conducted as if they were verification experiments, the idea that ubH2A may increase with age cannot be deduced from the proteome and ubiquitylated proteome data. The percentages of remaining ¹⁵N labels on day 60 are the same for H2A and ubH2A@K118, which means that for histone H2A, ubiquitination does not further increase the half-life of the protein. Therefore, increase of H2A K118 ubiquitylation with age is a chance discovery.

Second, I don't think the data presented in this manuscript support the conclusion that reducing ubH2A promotes adult healthy lifespan in *Drosophila*. Null mutations of *Sce* and *Su(z)2* disrupt PRC1, which is critical for chromatin silencing in addition to H2A monoubiquitization. No data were presented to show that these mutations only affected H2A K118 monoubiquitization.

In sum, this is a high-quality study, but the authors shall not make bold claims not supported by data. I recommend major revision.

Reviewer #2 (Remarks to the Author):

Yang et al. performed a pulsed metabolic labeling using N15 to identify long-lived proteins throughout the lifespan of flies. They analysed four different tissues (head, muscle and testis) and identified groups of proteins that retain at least 10% of the original N15 label after 60 days of life. Some of these proteins recapitulate the findings of a previous study by Toyama et al. that employed a similar strategy to study long-lived proteins in rat brain (Toyama et al. Cell 2013). In addition, the authors of this study performed an analysis of ubiquitylated peptides by applying an antibody enrichment strategy coupled to mass spectrometry. One of their key findings is that ubiquitylated peptides retain more N15 label as compared to the general sample suggesting that long-lived proteins accumulate ubiquitylation during aging. Among the ubiquitylated peptides that show higher retention of N15 label, they focus on peptides deriving from histones H2A and H2B. The authors validated the increased levels of ubiquitylated H2B by Western Blot and, interestingly, they show similar age-dependent patterns in mice (brain) and humans (cortex). Finally, they show that reducing the levels of modified H2B via the knockout of two enzymes responsible for this modification is sufficient to extend lifespan in flies. They propose that the lifespan extension is mediated by de-repression of detoxifying enzymes including glutathione-related pathways,

There is a lot of ongoing research linking epigenetic changes (either DNA methylation and histone modifications) to the aging process. Yang et al. identify the increased level of ubiquitylation of H2A as a novel marker of aging in multiple species, and show that by genetically reducing its level in flies is sufficient to extend life span. These findings are extremely relevant and timely, however there are key points that need to be addressed for the manuscript to be considered for publication in Nature Communications.

Major points:

- Mass spectrometry experiments are well designed and performed, however they appear to have been performed only once. In order to support the claims of Figure 1 and 2, the reproducibility of the findings needs to be demonstrated by performing more replicate experiments.

- I am also not convinced by the evidence provided that ubiquitylated proteins show higher retention of the N15 labels. The analysis showed in Figure 2A is based on the number of PSMs assigned to either of the isotopologues, however this compares measurements performed on completely different matrices (total proteome and enriched sample for ubiquitylated peptides). The authors should rather compare peptide wise N15/N14 ratio distributions for both the experiments (total and enriched samples) to support their statement. This should be assessed in a tissue-by-tissue manner in order to explore differences between tissues.

- A major finding is the increased level of UbH2A/B across species. The preliminary data by Western Blot (Figure 3) look promising but need to be further corroborated by an independent validation of UbH2A/B increase by, for example, mass spectrometry.

- How general is the increase in UbH2A/B across tissues? More tissues should be analyzed for mouse and humans.

- My major concerns are regarding the lifespan experiments in *Drosophila*. The authors depleted two major enzymes responsible for the ubiquitylation of H2A in whole fly using CRISPR. The authors confirmed the deletion of the target genes, however they do not show validation of the knock-out at the protein level. Western blots from deletion mutants flies for Sce and Su(z)2, as well as other components of the PRC2 complex should be provided. The authors should also check the effect on other histone marks that are related to UbH2A, namely methylation on histone H3 at lysine 4.

- The authors performed whole fly knock-out, however they originally observed the increase of UbH2A in head samples. Would depleting Sce and Su(z)2 exclusively in one tissue and, perhaps, only at old age induce the same effect on lifespan. This reviewer is not a *Drosophila* expert, but these experiments should be doable using standard fly genetics tools, and they would greatly strengthen the impact of the manuscript.

- Another key point that should be addressed is whether the increase in UbH2A is occurring at specific sites in the genome or it is widespread. A ChIP-seq experiment in at least one tissue should be performed. This should be compared to the gene expression response induced by the depletion of Sce and Su(z)2. How does depleting rather pleiotropic histone modifiers generate this specific response?

- The gene expression profile induced by the depletion of Sce and Su(z)2 appears a typical stress-response profile, including increased expression of detoxifying enzymes and ER stress. The authors

should compare this profile to one induced by other lifespan extending interventions such as dietary restriction or other long lived mutants.

- Several key references are not cited. For example, Increased levels ubiquitylated H2B were previously found in yeast replicative aging <https://www.ncbi.nlm.nih.gov/m/pubmed/24025678/>. The authors should mention this paper. Similarly the authors speculate in the discussion regarding decreased activity of the Ubiquitin Proteasome System with aging. This has already been shown in fly, and the authors should discuss relevant literature: e.g., <https://www.ncbi.nlm.nih.gov/pubmed/17413001>

Minor points

- The authors should indicate protein and peptide level False Discovery Rate (FDR) in the mass spectrometry method session.

- Very limited information is provided regarding the assignment of ubiquitylation sites. What kinds of filters were applied for the sites identification? More details need to be provided to assess the quality of the data and for the reproducibility.

- There are a number of imprecise statements throughout the manuscript that should be corrected: (i) I have some issues with the usage of the term SILAM, since the authors do not really amino acids for metabolic labeling of proteins but N15. The authors should change this acronym to better reflect their experiment. (ii) When referring to previous work by Toyama et al., the authors sometimes uses the term 'murine', while the cited paper was performed in rats. These statements should be corrected. (iii) Page 5, the authors mention laminins as example of long lived proteins that were found in a previous in rat as well, and they state that 'laminin of the nuclear proteins are evolutionary conserved ELLPs'. I am not sure exactly what this sentence means, but I have the feeling that the authors are mixing laminins (components of the extracellular matrix) with lamins (components of the nuclear envelope). The authors should check this sentence and correct it accordingly.

Reviewer #3 (Remarks to the Author):

The manuscript by Lu Yang and colleagues asks a very interesting question: what happens to exceptionally stable proteins during animal aging? Different proteins have very different turnover rates: some are replaced as a matter of hours and some as a matter of days or longer. What happens to the exceptionally stable, long-lasting proteins as animals age has not been examined in detail.

The authors use isotope labeling and proteomic analysis to define the subset of fruit fly proteins that have exceptional stability, predominantly retaining the label for over 60 days. They focus on adult male tissues/structures: the head, muscle and testes and successfully identify a set highly stable proteins, which is enriched in several GO categories. The authors then focus on ubiquitylation. They find that old proteins are more ubiquitinated. This is presumably due to accumulated damage. Here they focus on the histone proteins, in particular on H2A, which they find is stable and whose ubiquitylation increases with age, in flies, mice and humans. They show that knockdown of the genes (*Sce* and *Su(z)2*) thought to be responsible for this modification can extend lifespan in the male fly.

The question addressed is very interesting. The data are of good quality and appear correctly analyzed. My biggest concern is the lack of clear connections between age (or aging), the increase in H2 ubiquitylation, *Sce* and *Su(z)2* and lifespan.

For example: Why is H2 ubiquitinated as fly get older? Is this the regulatory mono Ubq modification or is this a poly Ubq and a marker of damage? I believe the authors indicate it's regulatory but what is then the link to H2 being a stable protein. Are *Sce/Su(z)2* responsible for this increase? Is their activity increased with age? Does reducing their activity impact age-related changes in H2 ubiquitylation (or even proteome wide changes)? Additionally, among all the functions of *Sce/Su(z)2*, how can the authors be sure the extended lifespan is due to the impact on H2 ubiquitylation?

I think the authors would need to present additional data to clarify at least some of these points and strengthen the mechanistic insight presented in the paper.

Further concerns:

- 1) For lifespans in Figure 4a: please give exact number of dead/censored flies. Please show the *Su(z)2* mutant alone (not in combination with *Sce*). Please show at least one more, completely independent experimental repeat. If possible, please include data on females (even if negative).

- 2) For oxidative stress assays: please test if there are any confounding differences in feeding rates between the genotypes. Ideally, there should be at least two repeats of all phenotypic tests shown.
- 3) There are a lot of proteomics data presented. It would be good for the authors to include some measure of false discovery rate with their analyses. Same for RNA-Seq.
- 4) Can the half-life of the exceptionally stable proteins be estimated?
- 5) Please clarify the exact modification that is being detected in the western blots presented – on the figure or in figure caption.
- 6) The use of “long-lived” for both proteins and flies is a bit confusing. I would suggest to call the proteins “stable” rather than long-lived.
- 7) For fly genetics: please state explicitly how the deletions were tracked during backcrossing. Also, were the experimental flies standardized for all cytoplasmic, maternally inherited genetic information?

Reviewer #4 (Remarks to the Author):

In the manuscript titled “Ubiquitylome study identifies epigenetic modulation of ubH2A as an evolutionarily conserved aging hallmark”, the authors, Yang et al., reports a study that combines long-lived proteome with ubiquitylome analysis in the fruit fly *Drosophila melanogaster*. They found that long-lived proteins tend to contain more ubiquitylation. Furthermore, they identified ubH2A as a new mark for aging, not only in *Drosophila*, but also in mouse and human tissues. Genetically disrupt the related E3 ligases reduced ubH2A levels and extended lifespan in flies, likely through elevated stress responses.

Overall, this manuscript was clearly written; the experiments were thoughtfully designed and well executed. Most of the data presented supported their conclusions and were convincing. The main conclusions are very interesting and represent a significant advance in our understanding of the role of ubiquitylation, especially histone ubiquitylation, on aging regulation.

However, there are a couple of concerns on the authors' interpretation of the data presented in the manuscript. If these concerns are carefully addressed, the overall quality and readability of the manuscript will be much improved.

1) The authors showed an additive effect in lifespan and ubH2A for *Sce* and *Su(z)2* mutants in *Drosophila* (Fig 4a-b). They interpreted it simply as a dosage-dependent effect. This is likely true. However, this additive and dosage-dependent effect caused by combining two mutants is probably best interpreted as functional redundancy between the two gene products. In fact, such functional redundancy between *Sce* and *Su(z)2* has been implicated in previous studies, such as Prado et al. PLOS One, 2012.

2) In the GO analysis shown in Fig 4c, the glutathione category was the least significant one ($-\log_{10}P < 3$, x-axis label incorrect in this figure). Many other categories had more significant p values, including unfolded protein response ($-\log_{10}P$ about 3.5), which has already been implicated in aging regulation by many recent studies. Why is this category not further investigated? The results shown in Fig 4e is the least impressive data in this manuscript. It is not sufficient to explain either the longevity or the oxidative stress resistance observed by the *Sce* single mutant (Fig 4a and 5b). Could unfolded protein response play a role here? Maybe the combined effects of unfolded protein response and glutathione-based detoxification can better explain the observed longevity and stress resistance phenotypes? If the authors do not think that this is the case, evidence that rules out the involvement of unfolded protein response should be presented.

Response to the decision and review comments

First of all, we thank the Reviewers for their comments and suggestions. Below, italic fonts are Reviewers points, followed by our response in blue, detailing how we have revised the manuscript and added new experiments to address the concerns from the reviewers. Additional references are shown in the end. We hope that you will now find the manuscript acceptable for publication in *Nature Communications*.

We have addressed the Reviewers comments with extensive additional data. These key new data include:

1. To further support the evolutionarily conserved role of ubH2A as a biomarker for aging, we examined the dynamics of ubH2A level during aging using additional tissue types and from an additional mammalian species, Rhesus macaque (*Macaca mulatta*). Our data now demonstrate that abundances in ubH2A increase similarly during aging in mouse brain, heart, and *Macaca mulatta* brain (Fig 3d,f,h).
2. To independently validate the changes of ubH2A during aging, we used mass spectrometry to quantitatively measure the change of ubH2A from fly head, mouse brain, mouse heart, as well as *Macaca mulatta* brain samples. The results support our findings by western blotting (Fig 3c,e,g,i).
3. We have included the data showing that *Sce*^{c244}, *Su(z)2*^{c433} female mutant flies have an extended lifespan, a longevity phenotype as observed in male mutant flies (Supplementary Fig 5).
4. As a control, we demonstrate that *Sce*^{c244}, *Su(z)2*^{c433} deficient flies have a specific effect on the level of ubH2A, but not other epigenetic markings (Supplementary Fig 7).

Reviewers' comments:

Reviewer #1

The authors of this study used state-of-the-art proteomics technology to characterize long-lived proteins in Drosophila. Head, muscle, and testis samples were analyzed on adult days 5, 30, and 60. The flies were shifted from 15N-labeled food to unlabeled food on adult day 5. The authors analyzed the whole proteome and the ubiquitylated proteome in parallel and quantified for each peptide the percentage of remaining 15N labels. Proteins with a longer half-life will retain a high level of 15N labels. A total of 149 extremely long-lived proteins (ELLPs) were found by taking the overlap between the head and the muscle data. There is a good agreement between the fly ELLPs from this study and the mouse ELLPs from previous studies. Fly specific ELLPs were also identified. One very interesting discovery is that ubiquitylated peptides tend to retain a higher level of 15N labels than peptides not ubiquitylated, suggesting that ubiquitylated proteins tend to have a slower turnover rate or a longer half-life. This is

unexpected but very intriguing, and it provides insight to the metabolism of the aging proteome. I think this is an important discovery and the data sets generated in this study will be an asset to the aging research community.

We thank the Reviewer for highlighting the significance of the work and the impact to the field.

However, this manuscript is not without flaws. Two major ones:

First, that ubH2A is a highly conserved hallmark of aging is a premature conclusion. Yes, it is exciting to observe an increase of ubH2A in aged Drosophila, mice, and human, but to call it a hallmark of aging, one needs to examine it under different genetic or environmental conditions that alter lifespan. For example, between wild-type and long-lived mutant flies of the same age, is the ubH2A level higher in the former?

Response: We appreciate the issue raised by the Reviewer. Given the scope of our current study, we have now changed the title into: Ubiquitylome study identifies epigenetic modulation of ubH2A as an evolutionarily conserved aging biomarker. To strengthen this statement, we further examined mouse brain, heart, and the brains of an additional mammalian species, *Macaca mulatta* monkey. Our finding demonstrates that the level of increased ubH2A is a highly evolutionarily conserved feature of normal aging of different species (now in Fig 3 of the revised manuscript).

Additionally, I like to point out that an increase of ubH2A with age is an incidental discovery. Although the WB experiments were conducted as if they were verification experiments, the idea that ubH2A may increase with age cannot be deduced from the proteome and ubiquitylated proteome data. The percentages of remaining ¹⁵N labels on day 60 are the same for H2A and ubH2A@K118, which means that for histone H2A, ubiquitination does not further increase the half-life of the protein. Therefore, increase of H2A K118 ubiquitylation with age is a chance discovery.

Response: We agree with the Reviewer that the age-onset increase of ubH2A cannot be deduced solely based on the proteomic datasets collected on the day 60 of aged flies. Our quantitative measurements that profiled the interface between ubiquitylation and aging proteome note a significant enrichment of ubiquitylated histone 2A. We now provide the data to demonstrate the increase of ubH2A by both western blotting and mass spectrometry, as shown in the revised manuscript (Fig 3b-c of the revised manuscript).

At day 60 of aged flies, our data showed that while ¹⁵N-labeled old forms accounted for 78% of H2A, they represented 82% of the ubiquitylated H2A. Although the ratio of ¹⁵N/¹⁴N is slightly increased for ubH2A, our data does not suggest that ubiquitylation contributes to the stability of histone protein in aging context. In this case, ubiquitylation on histone protein may be linked to epigenetic and chromatin modifications which should be examined by further studies. We have discussed this in

the first paragraph of the Discussion (Line 214-220, Page 9).

Second, I don't think the data presented in this manuscript support the conclusion that reducing ubH2A promotes adult healthy lifespan in Drosophila. Null mutations of Sce and Su(z)2 disrupt PRC1, which is critical for chromatin silencing in addition to H2A monoubiquitization. No data were presented to show that these mutations only affected H2A K118 monoubiquitization.

Response: Although our additional control data demonstrates that the heterozygous mutation of *Sce* and *Su(z)2* genes does not affect other histone markings, we have toned down our conclusion to indicate that our discovery provides ubH2A as an aging biomarker. We agree that the functional significance of ubH2A in aging should be examined by further studies.

Our genetic evidence was based on studying heterozygous mutation of *Sce* and *Su(z)2* genes, in which ubH2A modification, despite at a reduced level, remained operative in mutant animals. Indeed, Null mutations of *Sce* and *Su(z)2* cause pre-adult lethality, such that their age-associated activity cannot be addressed. In our recent report, we were able to characterize adult, age-related function of PRC2 factors by using similar heterozygote-based deficiency approach¹ (PMID: 29809154). In the revised manuscript, we have carefully labeled mutation genotype as heterozygote in the figure panels, figure legends, and methods.

In the revised manuscript, we examined other epigenetic modifications, which remained unchanged in the *Sce* and *Su(z)2* heterozygous double mutants, except for ubH2A (now in Supplementary Fig 7 of the revised manuscript). These data support the notion that *Sce* and *Su(z)2* heterozygote double mutation specifically affects the level of ubH2A.

Reviewer #2

Yang et al. performed a pulsed metabolic labeling using N15 to identify long-lived proteins throughout the lifespan of flies. They analysed four different tissues (head, muscle and testis) and identified groups of proteins that retain at least 10% of the original N15 label after 60 days of life. Some of this proteins recapitulate the findings of a previous study by Toyama et al. that employed a similar strategy to study long lived proteins in rat brain (Toyama et al . Cell 2013). In addition, the authors of this study performed an analysis of ubiquitylated peptides by applying an antibody enrichment strategy coupled to mass spectrometry. One of their key findings is that ubiquitylated peptides retain more N15 labeled as compared to the general sample suggesting that long-lived proteins accumulate ubiquitylation during aging. Among the ubiquitylated peptides that show higher retention of N15 label, they focus on peptides deriving from histones H2A and H2B. The authors validated the increased levels of ubiquitylated H2B by Western Blot and, interestingly, they show similar age-dependent patterns in mice (brain) and humans (cortex). Finally, they show that reducing the levels of modified H2B via the knockout of two enzymes responsible for

this modification is sufficient to extend lifespan in flies. They propose that the lifespan extension is mediated by de-repression of detoxifying enzymes including glutathione-related pathways,

There is a lot of ongoing research linking epigenetic changes (either DNA methylation and histone modifications) to the aging process. Yang et al. identify the increased level of ubiquitylation of H2A as a novel marker of aging in multiple species, and show that by genetically reducing its level in flies is sufficient to extend life span. These findings are extremely relevant and timely, however there are key points that need to be addressed for the manuscript to be considered for publication in Nature Communications.

We thank this reviewer for appreciating the significance and impact of our study. We have worked hard to provide new data to address all of the concerns.

Major points:

- Mass spectrometry experiments are well designed and performed, however they appear to have been performed only once. In order to support the claims of Figure 1 and 2, the reproducibility of the findings needs to be demonstrated by performing more replicate experiments.

Response: We apologize for the lack of clarity. The analyses for aging proteomes have been independently repeated for three times, while ubiquitylome analyses have been repeated for two times. In addition, we have conducted parallel mass spec analyses on multiple tissue types (head and muscle) and from different species including fly, mouse, monkey, and human. We have added this information in the “methods” section (Line 276, Page 11).

- *I am also not convinced by the evidence provided that ubiquitylated proteins show higher retention of the N15 labels. The analysis showed in Figure 2A is based on the number of PSMs assigned to either of the isotopologues, however this compares measurements performed on completely different matrices (total proteome and enriched sample for ubiquitylated peptides). The authors should rather compare peptide wise N15/N14 ratio distributions for both the experiments (total and enriched samples) to support their statement. This should be assessed in a tissue-by-tissue manner in order to explore differences between tissues.*

Response: We thank the Reviewer for raising this question. PSM is a useful indicator to estimate an overall labeling efficiency. The data shown in Fig 2b is derived from XIC ratio-based ¹⁵N/¹⁴N quantification. To address this concern, we analyzed the fly muscle samples in the same manner, and obtained a similar result (now in Supplementary Fig 3 of the revised manuscript).

- *A major finding is the increased level of UbH2A/B across species. The preliminary data by Western Blot (Figure 3) look promising but need to be further corroborated by an independent validation of UbH2A/B increase by, for example, mass spectrometry.*

Response: As suggested by the Reviewer, we have confirmed this result using

targeted mass spectrometry, as now shown in Figure 3c,e,g,i of the revised manuscript.

- *How general is the increase in UbH2A/B across tissues? More tissues should be analyzed for mouse and humans.*

Response: We have extended our analysis into mouse brain, mouse heart, *Macaca mulatta* brain samples by using both western blotting and mass spectrometry, as now shown in Figure 3c-i of the revised manuscript. The results support the increased ubH2A as an evolutionarily conserved biomarker in aging.

- *My major concerns are regarding the lifespan experiments in Drosophila. The authors depleted two major enzymes responsible for the ubiquitylation of H2A in whole fly using CRISPR. The authors confirmed the deletion of the target genes, however they do not show validation of the knock-out at the protein level. Western blots from deletion mutants flies for *Scd* and *Su(z)2*, as well as other components of the PRC2 complex should be provided. The authors should also check the effect on other histone marks that are related to UbH2A, namely methylation on histone H3 at lysine 4.*

Response: We thank the Reviewer for raising this question. We would like to clarify that our genetic evidence was based on studying heterozygous mutation of *Scd* and *Su(z)2* genes. In the revised manuscript, we have carefully labeled mutation genotype as heterozygote in the figure panels, figure legends, and methods.

As for many *Drosophila* proteins, we could not find good antibodies to detect *Scd* and *Su(z)2* proteins. However, we presented the data that transcription of *Scd* and *Su(z)2* genes was decreased in the heterozygote double mutants (Supplementary Fig 4c). Moreover, the ubH2A level was indeed reduced as shown by western blotting (Fig 4b), indicating that the activity of *Scd* and *Su(z)2* is downregulated in the heterozygote double mutants. Importantly, we have added new data to demonstrate that the *Scd*^{c244}, *Su(z)2*^{c433} heterozygote double mutants specifically affect the level of ubH2A, but not other epigenetic markings (now in Supplementary Fig 7 of the revised manuscript).

- *The authors performed whole fly knock-out, however they originally observed the increase of UbH2A in head samples. Would depleting *Scd* and *Su(z)2* exclusively in one tissue and, perhaps, only at old age induce the same effect on lifespan. This reviewer is not a Drosophila expert, but these experiments should be doable using standard fly genetics tools, and they would greatly strengthen the impact of the manuscript.*

Response: We had explored the possibility of knockout *Scd* and *Su(z)2* in fly heads. Likely due to the role of complete knockout of *Scd* and *Su(z)2* in regulating PRC1 and PRC2 expression, *geneswitch-GAL4* flies alone had a deleterious effect on adult life. Since our experiments used double heterozygous *Scd* and *Su(z)2* mutants, it is difficult to construct such tissue specific double het-mutants. In addition, we have toned down our conclusion to say that our study demonstrates ubH2A as a biomarker for aging.

The functional significance of our finding will need to be addressed by future studies.

- Another key point that should be addressed is whether the increase in UbH2A is occurring at specific sites in the genome or it is widespread. A ChIP-seq experiment in at least one tissue should be performed. This should be compared to the gene expression response induced by the depletion of *Sce* and *Su(z)2*. How does depleting rather pleiotropic histone modifiers generate this specific response?

Response: This question is important to address the functional significance of ubH2A. We have tried to address this question by conducting ubH2A ChIP-seq experiment. We have tried two antibodies, #8240 by CST and 05-678 by Millipore. Unfortunately, these antibodies, though good for western immunoblot, show poor specificity for ChIP study using animal tissue samples as which cannot determine unique peak enrichment of IP from that of Input (Figure R1). Since we now limit our conclusion on ubH2A as a biomarker for aging, such experiments should be done in future by first developing specific, ChIP-grade antibody. Because we have been unable to do this, we have referred to our data in a way as to not overstate the findings, and highlight the interesting future experiments possible, on lines 251-254 of the revised manuscript.

Figure R1. Genome browser view shows the ChIP-seq profile of ubH2A using antibodies generated by CST and Millipore. Head tissues from day 3 flies were used. ChIP experiments were done as previously described¹ (PMID: 29809154).

- The gene expression profile induced by the depletion of *Sce* and *Su(z)2* appears a typical stress-response profile, including increased expression of detoxifying enzymes and ER stress. The authors should compare this profile to one induced by other lifespan extending interventions such as dietary restriction or other long lived mutants.

Response: Thank you for this suggestion. We examined the RNAseq data for long-lived *piwi* deficiency mutant flies. As shown in Figure R2, oxidative stress pathways have not been upregulated in *piwi* mutants.

Figure R2. mRNAseq of long-lived *piwi* deficiency mutant flies.

- Several key references are not cited. For example, Increased levels ubiquitylated H2B were previously found in yeast replicative aging <https://www.ncbi.nlm.nih.gov/m/pubmed/24025678/>. The authors should mention this paper. Similarly the authors speculate in the discussion regarding decreased activity of the Ubiquitin Proteasome System with aging. This has already been shown in fly, and the authors should discuss relevant literature: e.g., <https://www.ncbi.nlm.nih.gov/pubmed/17413001>

Response: We have added these key references (Line 239, 249 Page 9-10).

Minor points

- The authors should indicate protein and peptide level False Discovery Rate (FDR) in the mass spectrometry method session.

Response: The protein FDR was controlled at 1% for long-lived proteome profiling and ubiquitylome datasets. We have added this information (Line 328, Page 13).

- Very limited information is provided regarding the assignment of ubiquitylation sites. What kinds of filters were applied for the sites identification? More details need to be provided to assess the quality of the data and for the reproducibility.

Response: With a filter of protein FDR of 1%, the corresponding FDR for ubiquitinated peptide identifications is approximately 0.3% (Line 328, Page 13), and the confidence score for each identification is greater than 90%.

- There are a number of imprecise statements throughout the manuscript that should be corrected: (i) I have some issues with the usage of the term SILAM, since the authors do not really amino acids for metabolic labeling of proteins but ^{15}N . The authors should change this acronym to better reflect their experiment.

Response: SILAM stands for “stable isotope labeling in mammals”, in which the ^{15}N isotope is used. We agree with Reviewer that this nomenclature may not be appropriate in this study. In the revised manuscript, we have changed “SILAM” to

¹⁵N.

(ii) When referring to previous work by Toyama et al., the authors sometimes uses the term 'murine', while the cited paper was performed in rats. These statements should be corrected.

Response: Done.

(iii) Page 5, the authors mention laminins as example of long lived proteins that were found in a previous in rat as well, and they state that 'laminin of the nuclear proteins are evolutionary conserved ELLPs'. I am not sure exactly what this sentence means, but I have the feeling that the authors are mixing laminins (components of the extracellular matrix) with lamins (components of the nuclear envelope). The authors should check this sentence and correct it accordingly.

Response: We have edited this sentence (Line 109, Page 5).

Reviewer #3

The manuscript by Lu Yang and colleagues asks a very interesting question: what happens to exceptionally stable proteins during animal aging? Different proteins have very different turnover rates: some are replaced as a matter of hours and some as a matter of days or longer. What happens to the exceptionally stable, long-lasting proteins as animals age has not been examined in detail.

*The authors use isotope labeling and proteomic analysis to define the subset of fruit fly proteins that have exceptional stability, predominantly retaining the label for over 60 days. They focus on adult male tissues/structures: the head, muscle and testes and successfully identify a set highly stable proteins, which is enriched in several GO categories. The authors then focus on ubiquitylation. They find that old protein are more ubiquitinated. This is presumably due to accumulated damage. Here they focus on the histone proteins, in particular on H2A, which they find is stable and whose ubiquitylation increases with age, in flies, mice and humans. They show that knockdown of the genes (*Sce* and *Su(z)2*) thought to be responsible for this modification can extend lifespan in the male fly. The question addressed is very interesting. The data are of good quality and appear correctly analyzed.*

Thank you for appreciating the significance and impact of our study.

*My biggest concern is the lack of clear connections between age (or aging), the increase in H2 ubiquitylation, *Sce* and *Su(z)2* and lifespan. For example: Why is H2 ubiquitylated as fly get older? Is this the regulatory mono Ubq modification or is this a poly Ubq and a marker of damage? I believe the authors indicate it's regulatory but what is then the link to H2 being a stable protein. Are *Sce/Su(z)2* responsible for this increase? Is their activity increased with age? Does reducing their activity impact age-related changes in H2 ubiquitylation (or even proteome wide changes)? Additionally, among all the functions of *Sce/Su(z)2*, how can the authors be sure the*

extended lifespan is due to the impact on H2 ubiquitylation? I think the authors would need to present additional data to clarify at least some of these points and strengthen the mechanistic insight presented in the paper.

Response: We thank the Reviewer for appreciating the interest in the mechanism, and have attempted to address as many of their points as we could in this Revision.

To differentiate mono vs. poly ubiquitylation, western immunoblot with specific primary antibody allows the detection of mono ubiquitylated H2A of expected protein size. Previous findings have indicated that *Sce* and *Su(z)2* are components of the Polycomb Repressive Complex 1 (PRC1) that specifically modulates the mono-ubiquitylation of histone proteins²⁻⁷ (PMID: 430507, 2612506, 3253035, 16359901, 15386022, 15147763). Consistently, our current genetic evidence demonstrate that *Sce* and *Su(z)2* heterozygous mutation have reduced ubH2A. Nonetheless, to address the Reviewer's important concern regarding the specific link between *Sce* and *Su(z)2* and ubH2A, we have included additional data to indicate the specificity of *Sce* and *Su(z)2* mutants in the level of ubH2A but not on other histone markings (Supplementary Fig 7 of revised manuscript). Our results come from many collective observations. It is noteworthy that life-benefits, including lifespan and age-associated phenotypes (including locomotion and stress sensitivity) can be obtained by *Sce* single mutants, which can be further enhanced by *Sce* and *Su(z)2* double mutants, likely reflecting a dose-dependent or redundant effect. We also note that such assays are not definitive. We have however, in our writing tried to be clear to not overstate our findings. To avoid overstating, we have revised this manuscript to conclude that ubH2A as a biomarker of aging, and thus, many important functional questions should be addressed by future experiments.

Further concerns:

*1) For lifespans in Figure 4a: please give exact number of dead/censored flies. Please show the *Su(z)2* mutant alone (not in combination with *Sce*). Please show at least one more, completely independent experimental repeat. If possible, please include data on females (even if negative).*

Response: We have included the number of flies used in the related figure legends (Fig 4a, 5b, and Supplementary Fig 5a-b, 6, 8). We have added repeated lifespan data as shown in Supplementary Fig 5b and 6 of revised manuscript. As suggested by the Reviewer, we have added the lifespan data of *Su(z)2* mutant (Fig4a, Supplementary Fig6). We have additionally tested the lifespan in female *Sce*^{c244}, *Su(z)2*^{c433} double mutants as now in Supplementary Fig 5a-b of revised manuscript.

2) For oxidative stress assays: please test if there are any confounding differences in feeding rates between the genotypes. Ideally, there should be at least two repeats of all phenotypic tests shown.

Response: As indicated by food coloring in the gut, we found no difference in food intake between genotypes. We have added independent oxidative stress test as shown in Supplementary Fig 8 of revised manuscript.

3) *There are a lot of proteomics data presented. It would be good for the authors to include some measure of false discovery rate with their analyses. Same for RNA-Seq.*

Response: The protein FDR was controlled at 1% for long-lived proteome profiling and ubiquitylome datasets. We have added this information (Line 328, Page 13). For RNA-Seq, we yielded significant differential expression with the cutoff: adj p (p adjusted value) <0.05. And p value adjusted for multiple testing with Benjamini-Hochberg procedure, which controls false discovery rate (FDR=10%, the default setting of DESeq2)⁸ (PMID: 20979621).

4) *Can the half-life of the exceptionally stable proteins be estimated?*

Response: The half-life of a stable protein can only be roughly estimated. For precise measurement, a mathematic modeling and a series of sampling time points are needed. Our current datasets were collected from three time points with age, which could be used to profile proteins that have limited vs. quick turnover during adult lifespan.

5) *Please clarify the exact modification that is being detected in the western blots presented – on the figure or in figure caption.*

Response: We have added this information in the figures and figure legends (Fig 3b,d,f,h,j).

6) *The use of “long-lived” for both proteins and flies is a bit confusing. I would suggest to call the proteins “stable” rather than long-lived.*

Response: We thank the Reviewer to point out this. The terms “long-lived proteins” and “extremely long-lived proteins” (ELLPs) have been widely used by many previous literatures⁹⁻¹² (PMID: 23993091, 22300851, 19167330, 30552100). We suggest keeping this term for the description of proteins with limited turnover in the manuscript. To avoid confusion, we use “extended longevity” to describe fly mutants that live longer than WT.

7) *For fly genetics: please state explicitly how the deletions were tracked during backcrossing. Also, were the experimental flies standardized for all cytoplasmic, maternally inherited genetic information?*

Response: We apologize for the lack of clarity. The CRISPR-Cas9 mutagenesis and background clearance were done as previously described¹ (PMID: 29809154). To screen for flies that carries the deletion, single fly PCR analysis can be done using screen primers for *Scd*^{c244}

(F:5'-GTCGCTGTACGAGCTGCAGCGCAAG-3';R:5'-CGATGTGCCCGAATGACCCGGATCC-3') and *Su(z)2*^{c433} (F:5'-CGATCCAACCACTGTGGATTACTG-3'; R: 5'-GTAGGCAG TACCTCATCCTTGTAG-3'), respectively. The PCR results have now been shown in Supplementary Fig 4c of the revised manuscript. The virgin females carrying the deletion were backcrossed into WT (5905) male flies for five consecutive generations for a uniform genetic background, to mitigate background effects. We now provide more details in the Methods.

Reviewer #4 (Remarks to the Author):

*In the manuscript titled “Ubiquitylome study identifies epigenetic modulation of ubH2A as an evolutionarily conserved aging hallmark”, the authors, Yang et al., reports a study that combines long-lived proteome with ubiquitylome analysis in the fruit fly *Drosophila melanogaster*. They found that long-lived proteins tend to contain more ubiquitylation. Furthermore, they identified ubH2A as a new mark for aging, not only in *Drosophila*, but also in mouse and human tissues. Genetically disrupt the related E3 ligases reduced ubH2A levels and extended lifespan in flies, likely through elevated stress responses.*

Overall, this manuscript was clearly written; the experiments were thoughtfully designed and well executed. Most of the data presented supported their conclusions and were convincing. The main conclusions are very interesting and represent a significant advance in our understanding of the role of ubiquitylation, especially histone ubiquitylation, on aging regulation.

We appreciate the positive comments from this reviewer very much and have worked hard to address the comments.

However, there are a couple of concerns on the authors’ interpretation of the data presented in the manuscript. If these concerns are carefully addressed, the overall quality and readability of the manuscript will be much improved.

*1) The authors showed an additive effect in lifespan and ubH2A for *Sce* and *Su(z)2* mutants in *Drosophila* (Fig 4a-b). They interpreted it simply as a dosage-dependent effect. This is likely true. However, this additive and dosage-dependent effect caused by combining two mutants is probably best interpreted as functional redundancy between the two gene products. In fact, such functional redundancy between *Sce* and *Su(z)2* has been implicated in previous studies, such as Prado et al. *PLOS One*, 2012.*

Response: *We thank this reviewer for this important point. We agree and we have changed the text now read as: Since the effects of epigenetic genes may have functional redundancy¹³, we further analyzed adult survival using *Sce*^{c244}, *Su(z)2*^{c433} trans-heterozygous double mutants (line 166-168, page 6 of the revised manuscript).*

*2) In the GO analysis shown in Fig 4c, the glutathione category was the least significant one (-log10P <3, x-axis label incorrect in this figure). Many other categories had more significant p values, including unfolded protein response (-log10P about 3.5), which has already been implicated in aging regulation by many recent studies. Why is this category not further investigated? The results shown in Fig 4e is the least impressive data in this manuscript. It is not sufficient to explain either the longevity or the oxidative stress resistance observed by the *Sce* single mutant (Fig 4a and 5b). Could unfolded protein response play a role here? Maybe the combined effects of unfolded protein response and glutathione-based detoxification can better explain the observed longevity and stress resistance phenotypes? If the authors do not think that this is the case, evidence that rules out the involvement of unfolded protein*

response should be presented.

Response: We agree with the Reviewer that longevity phenotype may be contributed by combined activities of multiple pro-life pathways. We have changed text now read as: Pathway analysis revealed that glutathione-related pathway and unfolded protein response among others were significantly upregulated in double mutants as compared to age-matched WT animals (Fig. 4c). Specifically, the increase of GSTO1 has been previously implicated in oxidative stress response relevant to lifespan modulation in *C. elegans*¹⁴ (PMID: 25284791). qRT-PCR analysis confirmed that GSTO1 transcription had two-fold increase in double mutants relative to WT (Fig. 4d). We thus examined glutathione, an important indicator of the cellular redox state. Our data showed that GSH/(GSH+GSSG) ratio in mutants was significantly higher than that in WT (Fig. 4e), suggesting improved cellular redox potential in *Scec*²⁴⁴, *Su(z)2*^{c433} double mutants. Since some other genes also displayed altered expression between WT and age-matched mutants, we could not rule out the possibility that additional mechanisms might be involved. (Line 182, Page 7)

We again thank the Reviewers for their time and thoughtful consideration on our manuscript.

We hope with these changes the Editor and Reviewers will now find the manuscript acceptable for publication.

References

- 1 Ma, Z. *et al.* Epigenetic drift of H3K27me3 in aging links glycolysis to healthy longevity in *Drosophila*. *Elife* **7**, doi:10.7554/eLife.35368 (2018).
- 2 Goldknopf, I. L. & Busch, H. Isopeptide linkage between nonhistone and histone 2A polypeptides of chromosomal conjugate-protein A24. *Proceedings of the National Academy of Sciences of the United States of America* **74**, 864-868 (1977).
- 3 Lo, S. M., Ahuja, N. K. & Francis, N. J. Polycomb group protein Suppressor 2 of zeste is a functional homolog of Posterior Sex Combs. *Mol Cell Biol* **29**, 515-525, doi:10.1128/MCB.01044-08 (2009).
- 4 Gutierrez, L. *et al.* The role of the histone H2A ubiquitinase Sce in Polycomb repression. *Development* **139**, 117-127, doi:10.1242/dev.074450 (2012).
- 5 Gorfinkiel, N. *et al.* The *Drosophila* Polycomb group gene Sex combs extra encodes the ortholog of mammalian Ring1 proteins. *Mech Dev* **121**, 449-462, doi:10.1016/j.mod.2004.03.019 (2004).
- 6 Cao, R., Tsukada, Y. & Zhang, Y. Role of Bmi-1 and Ring1A in H2A ubiquitylation and Hox gene silencing. *Mol Cell* **20**, 845-854, doi:10.1016/j.molcel.2005.12.002 (2005).
- 7 Wang, H. *et al.* Role of histone H2A ubiquitination in Polycomb silencing. *Nature* **431**, 873-878, doi:10.1038/nature02985 (2004).
- 8 Anders, S. & Huber, W. Differential expression analysis for sequence count data. *Genome Biol* **11**, R106, doi:10.1186/gb-2010-11-10-r106 (2010).
- 9 Toyama, B. H. *et al.* Identification of long-lived proteins reveals exceptional stability of essential cellular structures. *Cell* **154**, 971-982, doi:10.1016/j.cell.2013.07.037 (2013).

- 10 Toyama, B. H. *et al.* Visualization of long-lived proteins reveals age mosaicism within nuclei of postmitotic cells. *J Cell Biol*, doi:10.1083/jcb.201809123 (2018).
- 11 Savas, J. N., Toyama, B. H., Xu, T., Yates, J. R., 3rd & Hetzer, M. W. Extremely long-lived nuclear pore proteins in the rat brain. *Science* **335**, 942, doi:10.1126/science.1217421 (2012).
- 12 D'Angelo, M. A., Raices, M., Panowski, S. H. & Hetzer, M. W. Age-dependent deterioration of nuclear pore complexes causes a loss of nuclear integrity in postmitotic cells. *Cell* **136**, 284-295, doi:10.1016/j.cell.2008.11.037 (2009).
- 13 Morillo Prado, J. R., Chen, X. & Fuller, M. T. Polycomb group genes Psc and Su(z)2 maintain somatic stem cell identity and activity in *Drosophila*. *PLoS one* **7**, e52892, doi:10.1371/journal.pone.0052892 (2012).
- 14 Schieber, M. & Chandel, N. S. TOR signaling couples oxygen sensing to lifespan in *C. elegans*. *Cell reports* **9**, 9-15, doi:10.1016/j.celrep.2014.08.075 (2014).

Reviewers' comments:

Reviewer #1 (Remarks to the Author):

In the revised version of “Ubiquitylome study identifies epigenetic modulation of ubH2A as an evolutionarily conserved aging biomarker” by Yang et al., the authors addressed well one of my two major concerns, but the other one—no sufficient evidence supporting the conclusion that reducing ubH2A promotes adult healthy lifespan in *Drosophila*—was neglected. That conclusion is a good example of a serious (but common, unfortunately!) logic error which confuses correlation with causation. To make it easier to see, I have simplified the authors’ reasoning as follows.

Fact: A causes x, y, z, and other effects

Conclusion 1: x causes z

Conclusion 2: x causes z through y

A = Mutation of *Sce* and *Su(z)2*

x = decreased ubH2A in older flies

y = increased GSTO1 expression

z = extension of healthy lifespan

To back up the claim that reducing ubH2A promotes adult healthy lifespan in *Drosophila*, which is a major conclusion of the manuscript, the authors must try different ways of reducing ubH2A in older flies and assay the following:

1. Does each increase healthy lifespan? (critical!)
2. Does each increased GSTO1 expression? (optional)
3. Does blocking GSTO1 expression by knock-down or knock-out suppress the lifespan extension effect? (optional)

Unless there is further evidence to lend support to it, that claim should be modified or removed.

Related, regarding the WB in Fig. 4b, please include Su(z)2c433/+ to match the survival curves in Fig. 4a. Also, how many times was the experiment repeated? Please show the repeat experiments in sup. info.

In the title the phrase “epigenetic modulation” is confusing. Yes, K118 ubiquitylation of H2A may speak for epigenetic, but what data speaks for modulation? Is it truly modulated? Or a passive change with increasing age? A more informative title may be: “Ubiquitylome study identifies an increase of ubH2A as an evolutionarily conserved aging biomarker.”

Reviewer #2 (Remarks to the Author):

The authors have done a good job in addressing the majority of my concerns.

There are still a couple of points that should be addressed before acceptance:

1. This reviewer appreciates the use of PRM to validate the Western Blot data for ubH2A. However the authors should provide more details on the experiments. For instance, it is not clear how the “young” and “old” samples compared were normalized. The authors should clearly state this in the methods.
2. In Figure S7, the authors probed a number of histone modifications in the mutant flies to demonstrate that Scec244, Su(z)2c433 heterozygote double mutants specifically affect the level of ubH2A. The authors provide quantification data from Western Blot but they do not show any of the original data. Please provide original uncropped version of the blots.

Reviewer #3 (Remarks to the Author):

The authors have essentially addressed my comments. There are three small adjustments that I would still like the authors to make:

- 1) The authors have now included the total number of flies in the lifespan assays in response to my comment (1). Please indicate how many were recoded as dead and how many were censored.

2) I think the feeding data authors refer to in response to my point (2) should be mentioned and shown in the manuscript.

3) The authors refer to the food they used as “standard” (line 265). Please give the exact composition of the food used. Apologies that I did not pick this up before.

Reviewer #4 (Remarks to the Author):

I have reviewed all the authors' responses to the review comments and evaluated the changes and many additions the authors have added to the manuscript. I feel that the authors have sufficiently addressed most of the concerns raised by reviewers and the manuscript has been improved as a result. I recommend this manuscript be accepted for publication in Nature Communications.

Response to the decision and review comments

First of all, we thank the Reviewers for their comments and suggestions. Below, italic fonts are Reviewers' points, followed by our response in blue, detailing how we have revised the manuscript. We hope that you will now find the manuscript acceptable for publication in *Nature Communications*.

We have addressed the Reviewers' comments with extensive additional data. These key new data include:

1. As suggested by Reviewer 1, we have changed the title of our manuscript into: Ubiquitylome study identifies an increase of ubH2A as an evolutionarily conserved aging biomarker.
2. To address the issue raised by Reviewer 1, we have toned down our conclusion to indicate that our discovery provides ubH2A as an aging biomarker. In the revised manuscript, we have changed the subtitle of the Result section into: Reducing ubH2A **couple**s with extended adult healthy lifespan in *Drosophila*.
3. As suggested by Reviewer 1, we have included western blotting result for *Su(z)2* single mutant (Fig 4b)
4. We have added the result of feeding rate assay for WT and *Sce*^{c244}, *Su(z)2*^{c433} deficient flies (Fig S4d).

Reviewers' comments:

Reviewer #1 (Remarks to the Author):

*In the revised version of “Ubiquitylome study identifies epigenetic modulation of ubH2A as an evolutionarily conserved aging biomarker” by Yang et al., the authors addressed well one of my two major concerns, but the other one—no sufficient evidence supporting the conclusion that reducing ubH2A promotes adult healthy lifespan in *Drosophila*—was neglected. That conclusion is a good example of a serious (but common, unfortunately!) logic error which confuses correlation with causation. To make it easier to see, I have simplified the authors' reasoning as follows.*

Fact: A causes x, y, z, and other effects

Conclusion 1: x causes z

Conclusion 2: x causes z through y

*A = Mutation of *Sce* and *Su(z)2**

x = decreased ubH2A in older flies

*y = increased *GSTO1* expression*

z = extension of healthy lifespan

To back up the claim that reducing ubH2A promotes adult healthy lifespan in

Drosophila, which is a major conclusion of the manuscript, the authors must try different ways of reducing ubH2A in older flies and assay the following:

1. Does each increase healthy lifespan? (critical!)
2. Does each increased GSTO1 expression? (optional)
3. Does blocking GSTO1 expression by knock-down or knock-out suppress the lifespan extension effect? (optional)

Unless there is further evidence to lend support to it, that claim should be modified or removed.

Response: We appreciate the issue raised by the Reviewer. We have toned down our conclusion to indicate that our discovery provides ubH2A as an aging biomarker. In the revised manuscript, we have changed the subtitle of the Result section into: Reducing ubH2A couples with extended adult healthy lifespan in *Drosophila* (Line 153, Page 6.). The mechanism that regulates ubH2A in aging will need to be investigated by future studies.

Related, regarding the WB in Fig. 4b, please include *Su(z)2c433/+* to match the survival curves in Fig. 4a. Also, how many times was the experiment repeated? Please show the repeat experiments in sup. info.

Response: As suggested by the Reviewer, we have provided Figure 4b with western blotting data from four fly lines. The lifespan experiment had two independent biological repeats (Figure 4a and Supplementary Figure 6). Western blotting analysis had four independent biological repeats (Source data for Figure 4b).

In the title the phrase “epigenetic modulation” is confusing. Yes, K118 ubiquitylation of H2A may speak for epigenetic, but what data speaks for modulation? Is it truly modulated? Or a passive change with increasing age? A more informative title may be: “Ubiquitylome study identifies an increase of ubH2A as an evolutionarily conserved aging biomarker.”

Response: We have changed the title as suggested by this Reviewer. Thanks.

Reviewer #2 (Remarks to the Author):

The authors have done a good job in addressing the majority of my concerns.

There are still a couple of points that should be addressed before acceptance:

1. *This reviewer appreciates the use of PRM to validated the Western Blot data for ubH2A. However the authors should provide more details on the experiments. For instance, it is not clear how the “young” and “old” samples compared were normalized. The authors should clearly state this in the methods.*

Response: We apologize for the lack of clarity. The young and old samples with equal amount of starting materials (protein content) were analyzed by PRM method. The

acquired MS data was analyzed by Skyline software. The summed MS intensities of y3 to y7 ions, representing the five most intense peaks in MS2, were used for quantification. The MS intensities between different samples were not further normalized. We have added more details in the Method section: “Protein identification and quantification” (Page 12)

2) *In Figure S7, the authors probed a number of histone modification in the mutant flies to demonstrate that Scec244, Su(z)2c433 heterozygote double mutants specifically affect the level of ubH2A. The authors provide quantification data from Western Blot but they do not show any of the original data. Please provide original uncropped version of the blots.*

Response: We have added western blotting result in the Supplementary Figure 7. The uncropped western blotting images can be found in the Source data file.

Reviewer #3 (Remarks to the Author):

The authors have essentially addressed my comments. There are three small adjustments that I would still like the authors to make:

1) *The authors have now included the total number of flies in the lifespan assays in response to my comment (1). Please indicate how many were recoded as dead and how many were censored.*

Response: We have provided these data in the legend of lifespan figure panels.

2) *I think the feeding data authors refer to in response to my point (2) should be mentioned and shown in the manuscript.*

Response: We have added the feeding data in the Supplementary Figure 4d.

3) *The authors refer to the food they used as “standard” (line 265). Please give the exact composition of the food used. Apologies that I did not pick this up before.*

Response: We apologize for the lack of clarity. The standard diet used for fly in our study was composed of 36 g/L sucrose, 38 g/L maltose, 22.5 g/L yeast, 5.4 g/L agar, 60 g/L maizena, 8.25 g/L soybean flour, 0.9 g/L sodium benzoate, 0.225 g/L methyl-p-hydroxybenzoate, and 6.18 ml/L propionic acid. We have added the food recipe in the Method section (Line 264, Page 10).

Reviewer #4 (Remarks to the Author):

I have reviewed all the authors' responses to the review comments and evaluated the changes and many additions the authors have added to the manuscript. I feel that the authors have sufficiently addressed most of the concerns raised by reviewers and the

manuscript has been improved as a result. I recommend this manuscript be accepted for publication in Nature Communications.

We thank the Reviewer very much.